# Nasopharyngeal carriage, antimicrobial susceptibility patterns, and associated factors of Gram-positive bacteria among children attending the outpatient department at the University of Gondar Comprehensive Specialized Hospital, Northwest Ethiopia

**Abebe Birhanu**[1]*, **Azanaw Amare**[1], **Mitkie Tigabie**[1], **Eden Getaneh**[1], **Muluneh Assefa**[1], **Tena Cherkos**[2], **Feleke Moges**[1]

1 Department of Medical Microbiology, School of Biomedical and Laboratory Sciences, College of Medicine and Health Sciences, University of Gondar, Gondar, Ethiopia, 2 Department of Medical Parasitology, School of Biomedical and Laboratory Sciences, College of Medicine and Health Sciences, University of Gondar, Gondar, Ethiopia

* Abebe.Birhanu@uog.edu.et, abebebir10@gmail.com

## Abstract

### Background

Gram-positive bacteria residing in the nasopharynx can lead to severe illnesses in children, such as otitis media, pneumonia, and meningitis. Despite the potential threat, there is a lack of comprehensive data regarding the carriage rates of these bacteria among children in outpatient departments in the study area.

### Objective

This study aimed to assess the nasopharyngeal carriage, antimicrobial resistance patterns, and associated factors of Gram-positive bacteria among children attending the outpatient department at the University of Gondar Comprehensive Specialized Hospital, Northwest Ethiopia.

### Methods

A hospital-based cross-sectional study was conducted from May 1, 2023, to August 30, 2023. A total of 424 nasopharyngeal swab samples were collected using sterile nasopharyngeal swabs, inoculated on Blood Agar and Mannitol Salt Agar plates, and identified through colony morphology, Gram stain, and biochemical tests. Antimicrobial susceptibility of the identified bacterial isolates was determined employing both the Kirby-Bauer and modified Kirby-Bauer methods. D-tests were conducted using clindamycin and erythromycin discs to detect inducible clindamycin resistance, while cefoxitin disc tests were utilized to ascertain methicillin resistance. Data entry was executed using Epi-Data version 4.6, and subsequent

**Data Availability Statement:** All relevant data are within the paper.

**Funding:** The author(s) received no specific funding for this work.

**Competing interests:** The authors have declared that no competing interests exist.

**Abbreviations:** ATCC, American Type Culture Collection; BAP, Blood Agar Plate; CLSI, Clinical and Laboratory Standards Institute; CoNS, Coagulase-Negative Staphylococcus species; MDR, Multi-drug Resistant; MHA, Muller-Hinton Agar; MRCoNS, Methicillin Resistant Coagulase-Negative Staphylococcus species; MRSA, Methicillin-Resistant *Staphylococcus aureus*; MSA, Mannitol Salt Agar; UoGCSH, University of Gondar Comprehensive Specialized Hospital.

analysis was performed utilizing SPSS version 25. Bivariable and multivariable logistic regression analyses were employed to identify associated factors. An adjusted odds ratio at a 95% confidence interval with a *P*-value of < 0.05 was considered statistically significant.

## Results

The overall nasopharyngeal carriage rate of Gram-positive bacteria was 296/424 (69.8%, 95% CI: 65.3–74.0). *Staphylococcus aureus* was the most prevalent 122/424 (28.8%), followed by *Streptococcus pneumoniae* 92/424 (21.7%). Methicillin resistance was observed in 19/122 (15.6%) of *S. aureus* and 3/60 (5%) of coagulase-negative staphylococcus (CoNS) species. Inducible clindamycin resistance was 10/122 (8.2%) in *S. aureus* and 4/53 (7.5%) in coagulase-negative staphylococcus species. Multidrug resistance was found in 146/296 (49.3%, 95% CI: 43.6–55.0) of the isolates. Associated factors with a bacterial carriage were large family size (AOR = 3.061, 95% CI: 1.595–5.874, *P* = 0.001), having siblings under five years old (AOR = 1.991, 95% CI: 1.196–3.313, *P* = 0.008), indoor cooking (AOR = 2.195, 95% CI: 1.275–3.778, *P* = 0.005), an illiterate mother (AOR = 3.639, 95% CI: 1.691–7.829, *P* = 0.001), and hospital visits (AOR = 2.690, 95% CI: 1.405–5.151, *P* = 0.003).

## Conclusion

The study found a high nasopharyngeal carriage of Gram-positive bacteria in outpatient children, including notable levels of methicillin-resistant *S. aureus* and multi-drug-resistant isolates. Clindamycin, rifampin, and erythromycin were the most effective antimicrobials for the tested isolates. Factors contributing to bacterial carriage include visits to healthcare facilities, larger family sizes, having younger siblings, maternal illiteracy, and indoor cooking. This emphasizes the need for methicillin-resistant *S. aureus* surveillance in pediatric outpatient settings and community health education, especially for children's guardians. Additionally, improving household ventilation by separating kitchens from sleeping areas and regular screening of younger siblings in healthcare environments were recommended to reduce bacterial transmission within family members. The study also called for studies with advanced procedures like minimum inhibitory concentration testing and molecular characterization to better comprehend the resistance patterns and genes in circulating bacteria.

## Introduction

The nasopharynx provides a habitat for bacteria, mainly Gram-positive ones, and faces selective pressure from antimicrobials and vaccinations, especially in children [1]. Gram-positive bacteria that have been documented to colonize the nasopharynx as microbiota include *S. pneumoniae*, *S. aureus*, coagulase-negative staphylococcus (CoNS) species, and *Streptococcus pyogenes* [2, 3]. However, in young children, carrying these bacteria in the nasopharynx significantly contributes to various local and systemic infections such as otitis media, sinusitis, pneumonia, bacteremia, and meningitis, as these bacteria can migrate to adjacent areas. Moreover, this bacterial carriage can result in autoinfection and facilitate the transmission of infections to others, even though it may initially remain asymptomatic [4].

*S. pneumoniae* commonly colonizes the nasopharynx and is responsible for various bacterial infections in children, including otitis media, pneumonia, bacteremia, and meningitis, which occur through mechanisms such as aspiration, bacteremia, and local spread [5]. In addition, *S. pyogenes* causes a spectrum of diseases in school-age children, ranging from pharyngitis to severe conditions like streptococcal toxic shock syndrome and rheumatic heart disease [6].

*S. aureus* is commonly found colonizing the human anterior nares and nasopharynx and is associated with healthcare-related infections in children, particularly surgical site infections and central line-associated bloodstream infections [7, 8]. Unfortunately, *S. aureus* has developed resistance to commonly prescribed antimicrobials such as ampicillin, ciprofloxacin, erythromycin, gentamicin, chloramphenicol, methicillin, and even vancomycin, which is typically reserved as a last resort for treating methicillin-resistant *S. aureus* (MRSA) infections [9, 10]. The rise of invasive community-acquired MRSA presents a significant health concern in children, with incidence rates increasing drastically in the world [11, 12].

Coagulase-negative staphylococcus species are commonly found as part of the normal microbiota on the skin, in the anterior nares, and on the mucous membranes of healthy individuals. Despite being generally harmless, CoNS can lead to serious internal infections such as endocarditis, septicemia, prosthetic joint infections, urinary tract infections, and bloodstream infections, particularly following medical device usage [13]. Coagulase-negative staphylococcus species also serve in transferring antimicrobial-resistant genes to other bacteria through horizontal gene transfer. For instance, methicillin-resistant CoNS (MRCoNS) have contributed to the emergence of MRSA by providing the methicillin resistance gene located on the staphylococcal cassette chromosome mec. This transfer is facilitated by the ecological coexistence of MRCoNS and methicillin-sensitive *S. aureus* in the same areas of the human body [14].

Children are more likely than adults to consistently carry bacteria in their upper respiratory tract. The rate of nasopharyngeal carriage varies widely between countries, with particularly high rates in Africa. These rates are notably high during childhood and tend to decrease with age. For example, the prevalence is 63.2% in children under five years old, 42.6% in those aged five to fifteen, and 28.0% in individuals over fifteen [2, 15]. Respiratory tract infections rank fourth among the causes of global fatalities, with 2,603,913 deaths attributed to them. These infections are among the most serious illnesses caused by infectious agents [16]. Previous studies have reported varying prevalence rates of nasopharyngeal bacterial carriage in children across different regions, ranging from 0.44% [17] to 76.2% [18] in Europe, 19.4% [19] to 34.5% [20] in other African countries, and 29.9% [3] to 75.1% [21] in Ethiopia.

In developing nations, the overuse of antimicrobials has disrupted microbiota and heightened antimicrobial resistance, resulting in more frequent occurrences of resistant strains such as MRSA, cephalosporin-resistant *S. pneumoniae*, and penicillin non-susceptible *S. pneumoniae*. This problem is further exacerbated by improper prescription practices, a lack of patient education, limited diagnostic resources, unauthorized sales of antimicrobials, weak drug regulatory systems, and the use of antimicrobials in animal farming [1, 22–25].

In Asia, studies have shown that 18% to 80% of nasopharyngeal *S. pneumoniae* isolates in children are multi-drug resistant (MDR), with 40% of these isolates showing resistance to penicillin [22, 26]. In Europe, macrolide resistance in *S. pyogenes* varies, being under 4% in some countries, 5% to 39% in others and the USA, and over 40% in Asian countries [27]. In Africa, 17% to 43% of nasopharyngeal *S. pneumoniae* isolates in children are MDR, with over half resistant to penicillin, and MRSA carriage ranges from 0.49% to 31% [8, 23, 24, 28–30].

Although limited studies about the nasopharyngeal carriage of antimicrobial-resistant bacteria in outpatient children have previously been reported in Ethiopia, with an MDR level ranging from 17.7% to 56% [2, 31], carriage and resistance to commonly used antimicrobials vary greatly over time [22, 25, 32] within the same geographic area, level of local health facility,

and population. Furthermore, this study addressed gaps left by prior research by offering data on the carriage of Gram-positive bacteria and their antimicrobial resistance patterns, including profiles for inducible clindamycin resistance, which were not covered in earlier studies. Moreover, it is also wise to monitor and update the antimicrobial susceptibility patterns of the isolates, as antimicrobial resistance is becoming a real public health concern.

## Materials and methods

### Study area and setting

The study was conducted at the University of Gondar Comprehensive Specialized Hospital (UoGCSH) in Gondar town, Amhara, Ethiopia. The UoGCSH, a major teaching hospital with 977 beds, and 29 wards and emergency rooms, has served nearly 13 million people in its surrounding area and neighboring regions. It provides a wide range of medical services, including internal medicine, surgery, obstetrics and gynecology, pediatrics, laboratory tests, eye care, physiotherapy, dental care, cervical health, psychiatry, dermatology, and drug supply. The hospital also offers various social services and has specialized units for tuberculosis, kala-azar, cancer treatment, fistula surgery, psychiatric and psychological treatment, palliative and rehabilitation services, adult and pediatric intensive care unit, and cataract surgery. Gondar town is located 750 km northwest of Addis Ababa and has an estimated population of about 487, 224 in 2023.

### Study design and period

A hospital-based cross-sectional study was conducted from May 1, 2023, to August 30, 2023.

### Study population

The source population and study population were all children who visited the pediatric department and all children who visited the selected outpatient department during the study period at UoGCSH, respectively. Children who visit the outpatient department for well-child care and are under the age of fifteen were included in the study. Children under 15 and those with a history of nasal surgery, respiratory infection, and recent antimicrobial use were excluded from the study.

### Sample size determination and sampling technique

A single population proportion formula was used to calculate a total sample size of 424 using a 50% proportion value. A systematic random sampling technique was used to enroll 424 study participants. About 20 children have gone to the outpatient unit or section on each working day, based on the 2022 records of the hospital. Briefly, 20 children x 20 working days per month for four months gives a total of 1600 children. The sampling interval was calculated as 1600/424 = 4. The first child on the first day of sample collection was selected by a lottery method. After that, every fourth child was requested to participate and offer his or her informed assent until a sufficient number of samples were attained.

$$n = \frac{(Z_{a/2})^2 * p(1-p)}{d^2}$$

Where;
n is the minimum sample size required
Zα/2 is the Z score at 95% confidence interval, which is 1.96
p is the prevalence, which is 0.5

d is the margin of error, which is 5%

Then n = $(1.96)^2$ x (0.5) x (1–0.5)/ $(0.05)^2$

n = 384.2~385

After accounting for a 10% non-response rate, the final sample size was calculated as (n = 385 + 38.5 = 424).

## Data collection

Socio-demographic and clinical information about the children was gathered through a structured questionnaire administered via face-to-face interviews with their parents or guardians. A trained laboratory technologist collected the data, and the analysis was carried out with the assistance of a senior biostatistician.

## Specimen collection and processing

Trained nurses collected a total of 424 nasopharyngeal specimens from each child using sterile plastic nasopharyngeal swabs (FA INC., Korea). The procedure involved tilting the child's head back and inserting the swab into the nostril, advancing it to the nasopharyngeal area until resistance was felt. The swab was then rotated 4–5 times in both clockwise and counterclockwise directions (180 degrees) and held in place for five seconds to ensure adequate specimen absorption [15, 33].

After collection, the swabs were placed in labeled tubes containing Amies transport medium (Biomark, India) and transported to the bacteriology laboratory at UoGCSH in a cooler with ice packs, ensuring delivery within two hours. If processing within this time frame was not possible, samples were stored at temperatures between -80°C and -20°C to preserve the bacterial isolates [16].

## Isolation and identification of Gram-positive bacterial isolates

The nasopharyngeal samples were then streaked with a sterile wire loop on Mannitol Salt Agar plates (MSA) (HiMedia, India) and Blood Agar plates (BAP) (HiMedia, India). The BAP was incubated in a candle jar at 37°C for 24–48 hours, while the MSA was incubated aerobically for 18–24 hours at 37°C. Pure young colonies from BAP and MSA were subcultured on BAP and Nutrient Agar plates (NAP) (HiMedia, India) for biochemical and antimicrobial susceptibility testing.

Bacterial isolates were characterized using standard microbiological methods, including colony morphology, Gram stain, and biochemical tests [34]. Gram-positive diplococci and alpha-hemolytic colonies were transferred by a wire loop from BAP to a new BAP to obtain pure colonies. An optochin disc (5 μg) was placed on the inoculated streak using sterile forceps, and the plate was incubated for 18 to 24 hours at 37°C in 5% $CO_2$. Strains that showed an inhibition zone of $\geq$ 14 mm around the optochin disc were considered susceptible. To confirm bile solubility, the isolates were tested with 2% sodium deoxycholate (bile salt) (Oxoid, UK). Finally, *S. pneumoniae* was identified based on being Gram-positive diplococci, alpha-hemolytic, optochin-sensitive, bile-soluble, and catalase-negative.

A Gram-positive coccus with yellow colonies on MSA was subcultured onto BAP and incubated aerobically at 37°C for 18 to 24 hours to observe hemolysis patterns. Small, golden-yellow colonies from MSA, often displaying beta-hemolysis on BAP, were tested using a coagulase test to distinguish *S. aureus* from another *Staphylococcus* species. Finally, *S. aureus* was regarded as a Gram-positive coccus, beta-hemolytic, catalase-positive, coagulase-positive, and fermented mannitol, while CoNS was identified as a coagulase-negative and mannitol non-fermenter with pink colonies on MSA [35].

A Gram-positive coccus with large, grayish, β-hemolytic colonies, displaying dome-shaped morphology with smooth or moist surfaces and clear margins, was subcultured onto BAP to ensure colony purity. Colonies from the new BAP were then subjected to bacitracin (0.04 U) and catalase tests, followed by incubation in a 5% $CO_2$ atmosphere at 35–37˚C for 18–24 hours. Strains displaying a zone of inhibition ≥ 15 mm around the bacitracin disc were classified as sensitive. Gram-positive cocci exhibiting beta-hemolysis, catalase negativity, and susceptibility to 0.04 U bacitracin discs were reported as *S. pyogenes* [36].

## Antimicrobial susceptibility testing

Concisely, 3–5 young pure colonies were picked and emulsified with 5 mL of normal saline, and their turbidity was measured using 0.5 McFarland standards [3]. To standardize the bacterial suspension preparation, 0.85% normal saline was used. If the suspension was more turbid than the 0.5 McFarland standard, it was diluted accordingly. Conversely, if the suspension was less turbid, additional pure colonies of the target bacteria were added to achieve the correct turbidity. To prevent overflow, a sterile cotton swab was gently rotated to the tube's interior wall after being dipped into the adjusted suspension. Modified Kirby-Bauer disk diffusion method on Mueller-Hinton agar plate (MHA) (Biomark, India) enriched with 5% sheep blood was used to determine the antimicrobial susceptibility patterns of *S. pneumoniae* and *S. pyogenes*, whereas only Kirby-Bauer disk diffusion method on MHA was used to determine the antimicrobial susceptibility patterns of *S. aureus* and CoNS using the lawn culture method. After being permitted to air dry for 15 minutes, the discs were placed aseptically on the prepared plates using sterile forceps. Subsequently, the plates were incubated (modified MHA in a 5% $CO_2$ environment) at 37˚C for 24 hours.

Following Clinical and Laboratory Standard Institute (CLSI) guidelines [37], Gram-positive isolates of *S. pneumoniae*, *S. aureus*, CoNS, and *S. pyogenes* were tested against a range of antimicrobials including oxacillin (1μg), erythromycin (15μg), doxycycline (30μg), azithromycin (15μg), gentamicin (10μg), penicillin (10U), cefoxitin (30μg), tetracycline (30μg), ceftriaxone (30μg), chloramphenicol (30μg), clindamycin (2μg), vancomycin (30μg), ciprofloxacin (5μg), trimethoprim-sulfamethoxazole (1.25/23.75μg), ampicillin (10μg), and rifampin (5μg). Results of antimicrobial susceptibility testing were interpreted as sensitive, intermediate, or resistant based on CLSI 2022 [37]. Moreover, bacterial isolates demonstrating resistance to three or more antimicrobials in distinct classes were categorized as MDR [38].

**Detection of penicillin, methicillin, and inducible clindamycin resistance.** Penicillin non-susceptibility of *S. pneumoniae*, MRSA, and MRCoNS isolates was determined using oxacillin (1μg) (≤ 19 mm), cefoxitin (30μg) (≤ 21mm), and cefoxitin (30μg) (≤ 24 mm), respectively. Based on CLSI 2022 guidelines, Gram-positive bacterial isolates that showed resistance to erythromycin (15μg) (≤ 13 mm, ≤ 15 mm, ≤ 15 mm) and susceptibility to clindamycin (2μg) with a flat (D-inhibition) zone (≥ 21 mm, ≥ 19 mm, ≥ 19 mm) to *S. aureus* and CoNS, *S. pneumoniae*, and *S. pyogenes*, respectively were considered inducible clindamycin resistance.

## Quality control

The culture media were prepared based on the instructions of the manufacturer, and bacterial culture positivity was ensured by following standard operating procedures. Before using the freshly made culture media, 5% of the prepared batch was incubated at 35–37˚C for the entire 24 hours to confirm its sterility. Standard reference bacterial strains of *S. aureus* (American Type Culture Collection (ATCC)-13812), *S. pyogenes* (ATCC 12696), and *S. pneumoniae* (ATCC 12977) were used as control strains.

## Data analysis

The data was checked for completeness and coded, followed by data entry into Epi-Data version 4.6 and data export and analysis using Statistical Package for Social Sciences (SPSS) version 25. The frequency and percentage of variables were determined using descriptive analysis. Using the binary logistic regression model, independent variables with a $P$-value $\leq 0.25$ in the bivariable analysis were subjected to multivariable analysis to simultaneously control for the possible confounding effects of these variables. An adjusted odds ratio with a $P$-value $< 0.05$ at the 95% confidence interval (CI) was considered statistically significant. Finally, the results were presented in words, graphs, and tables.

## Ethical approval

Ethical clearance was obtained from the Ethical Review Committee of the School of Biomedical and Laboratory Sciences, College of Medicine and Health Sciences, University of Gondar, with reference number SBMLS/485, dated April 4, 2023. A letter of permission was obtained from UoGCSH. Before data collection commenced, every child participant provided their assent, and written informed consent was received from their parents or guardians. All data and samples were confidentially handled and solely utilized for this study. Children who carried Gram-positive bacteria were reported to physicians for appropriate medical care.

## Results

### Sociodemographic characteristics of children

A total of 424 children aged 3 months to 15 years were recruited, with a 100% response rate. Of these, 228 (53.8%) were female. The median and interquartile range of them were 4 years and 2–9 years, respectively. More than half (251) (59.2%) and (240) (56.6%) of them lived in cities and shared a house with five or more family members together, respectively, while (203) (47.9%) lived in a house with one room (Table 1).

### Prevalence of Gram-positive bacteria

The overall culture positivity of the nasopharyngeal carriage of Gram-positive bacteria among children attending the outpatient department was 296/424 (69.8%) (95% CI: 65.3–74.0). Of these, the culture positivity of *S. aureus* was 122/424 (28.8%) (95% CI: 24.6–33.3%), followed by *S. pneumoniae* 92/424 (21.7%) (95% CI: 18.0–25.9%), CoNS 60/424 (14.2%) (95% CI: 11.1–17.8%), and *S. pyogenes* 22/424 (5.2%) (95% CI: 3.43–7.8%) (Fig 1).

### Antimicrobial susceptibility pattern

The tested isolates exhibited high resistance to penicillin 197/296 (66.6%), tetracycline 180/296 (60.8%), and trimethoprim-sulfamethoxazole 135/296 (49.3%). Specifically, *S. aureus* isolates showed high resistance to penicillin 120/122 (98.4%) and tetracycline 90/122 (73.8%), but low resistance to erythromycin 37/122 (30.3%), clindamycin 26/122 (21.3%), and rifampin 12/122 (9.8%). Methicillin resistance was also present in 19/122 (15.6%) and 3/60 (5%) of *S. aureus* and CoNS, respectively. Similarly, more resistance was observed by isolates of *S. pneumoniae* to tetracycline 40/92 (43.5%), trimethoprim-sulfamethoxazole 38/92 (41.3%), doxycycline 35/92 (38.0%), penicillin 32/92 (34.8%), and erythromycin 31/92 (33.7%). However, there was less resistance to clindamycin 14/92 (15.2%), rifampin 5/92 (5.4%), and vancomycin 0/92 (0%). Likewise, noticeable resistance was shown by *S. pyogenes* isolates to tetracycline 10/22 (45.5%) and erythromycin 6/22 (27.3%), despite low resistance to chloramphenicol 2/22 (9.1%), clindamycin 2/22 (9.1%), and vancomycin 2/22 (9.1%), ampicillin 22/22 (100%) (Table 2).

**Table 1. Socio-demographic characteristics of children attending the outpatient department at the UoGCSH, Northwest Ethiopia, from May 1, 2023, to August 30, 2023.**

| Characteristics of study participants | Category | Frequency N (%) | Nasopharyngeal carriage of bacteria | |
|---|---|---|---|---|
| | | | Yes N (%) | No N (%) |
| Age | <3 years | 150 (35.4) | 108 (72) | 42 (28) |
| | 3–5 years | 104 (24.5) | 63 (60.6) | 41 (39.4) |
| | 6–10 years | 94 (22.2) | 72 (76.6) | 22 (23.4) |
| | 11–15 years | 76 (17.9) | 53 (69.7) | 23 (30.3) |
| Sex | Male | 228 (53.8) | 163 (71.5) | 65 (28.5) |
| | Female | 196 (46.2) | 133 (67.9) | 63 (32.1) |
| Residence | Urban | 251 (59.2) | 174 (69.3) | 77 (30.7) |
| | Rural | 173 (40.8) | 122 (70.5) | 51 (29.5) |
| Mothers' educational status | Illiterate | 71 (16.8) | 38 (53.5) | 33 (46.5) |
| | Elementary completed | 110 (25.9) | 75 (68.2) | 35 (31.8) |
| | Secondary completed | 125 (29.5) | 96 (76.8) | 29 (23.2) |
| | Diploma and above | 118 (27.8) | 87 (73.7) | 31 (26.3) |
| Fathers' education status | Illiterate | 85 (20.1) | 55 (64.7) | 30 (35.3) |
| | Elementary completed | 120 (28.3) | 87 (72.5) | 33 (27.5) |
| | Secondary completed | 123 (29.0) | 89 (72.4) | 34 (27.6) |
| | Diploma and above | 96 (22.6) | 65 (67.7) | 31 (32.3) |
| Fathers' occupational status | Governmental | 161 (38.0) | 116 (72.0) | 45 (28.0) |
| | Merchant | 141 (33.2) | 102 (72.3) | 39 (27.7) |
| | Farmers | 45 (10.6) | 29 (64.4) | 16 (35.6) |
| | Daily labour | 77 (18.2) | 49 (63.6) | 28 (36.4) |
| Mothers' occupational status | Governmental | 142 (33.5) | 106 (74.6) | 36 (25.4) |
| | Merchant | 112 (26.4) | 78 (69.6) | 34 (30.4) |
| | Housewife | 109 (25.7) | 74 (67.9) | 35 (32.1) |
| | Daily labour | 61 (14.4) | 38 (62.3) | 23 (37.7) |
| Income of parents | < 500 | 80 (18.9) | 68 (85.0) | 12 (15.0) |
| | 5001–1000 | 87 (20.5) | 57 (65.5) | 30 (34.5) |
| | 1001–1500 | 117 (27.6) | 76 (65.0) | 41 (35.0) |
| | >1500 | 140 (33.0) | 95 (68.0) | 45 (32.0) |
| Family size | <5 | 184 (43.4) | 150 (81.5) | 34 (18.5) |
| | ≥5 | 240 (56.6) | 146 (60.8) | 94 (39.2) |
| Siblings < 5 years | Yes | 220 (51.9) | 142 (64.5) | 78 (35.5) |
| | No | 204 (48.1) | 154 (75.5) | 50 (24.5) |
| Siblings ≥ 5 years | Yes | 205 (48.3) | 145 (70.7) | 60 (29.3) |
| | No | 219 (51.7) | 151 (68.9) | 68 (31.1) |
| Number of rooms | < 2 | 203 (47.9) | 135 (66.5) | 68 (33.5) |
| | 3–4 | 122 (28.8) | 83 (68.0) | 39 (32.0) |
| | ≥ 5 | 99 (23.3) | 78 (78.8) | 21 (21.2) |

**Inducible clindamycin resistance.** Inducible clindamycin resistance of the tested isolates was 10/122 (8.2) and 4/53 (7.5) for *S. aureus* and CoNS, respectively, despite no inducible clindamycin resistance to *S. pneumoniae* 0/92 (0%) and *S. pyogenes* 0/22 (0%) (Table 3).

**Multi-drug resistance pattern.** A total of 15 antimicrobials from 11 classes (penicillins, glycopeptides, tetracyclines, aminoglycosides, phenicols, macrolides, cephalosporins, lincosamides, fluoroquinolones, folate pathway antagonists, and ansamycins) were used to determine

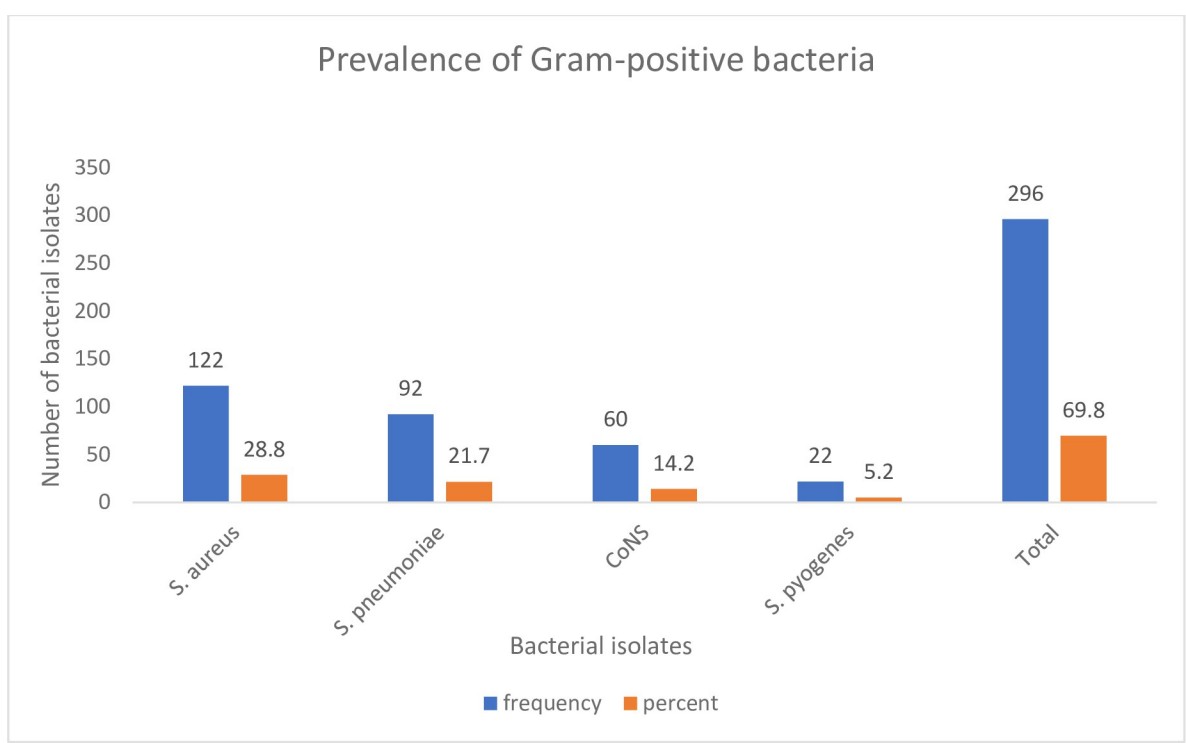

**Fig 1. Prevalence of Gram-positive bacteria isolated from nasopharyngeal samples of outpatient children at UoGCSH, Northwest Ethiopia, from May 1, 2023, to August 30, 2023.**

the patterns of MDR isolates. Two-hundred fifty-seven (86.8%) Gram-positive isolates showed resistance to at least one antimicrobial, while 39 (13.2%) of them were susceptible to the tested antimicrobials. The overall proportion of MDR was 146/296 (49.3%, 95% CI: 43.6–55.0). The predominant MDR isolate was *S. aureus* 23 (67.6%), followed by CoNS 24 (40%), and *S. pneumoniae* 36 (39.1%) (Table 4).

### Associated factors for nasopharyngeal carriage

In bivariate analysis, factors such as age, mothers' education, mothers' occupation, fathers' occupation, average income, family size, living with siblings ≤ 5 years, indoor cooking, number of rooms per house, having previous respiratory tract infection, history of hospitalization in the last three months, and visiting health care institutions reflected statistical association with COR at a *P*-value ≤ 0.25, and they were eligible for multivariate analysis (Table 5).

After entering into multivariate analysis, the most important risk factors for nasopharyngeal carriage of Gram-positive bacteria among children in this study were large family size (≥ 5 members) (AOR = 3.061, 95% CI: 1.595–5.874, *P* = 0.001), existence of siblings < 5 years (AOR = 1.991, 95% CI: 1.196–3.313, *P* = 0.008), indoor cooking (AOR = 2.195, 95% CI: 1.275–3.778, *P* = 0.005), having an illiterate mother (AOR = 3.639, 95% CI: 1.691–7.829, *P* = 0.001), and visiting health care institutions (AOR = 2.690, 95% CI: 1.405–5.151, *P* = 0.003). This study showed that children from large families, from families who practice indoor cooking, have siblings under the age of five, have an illiterate mother, and history of hospital visits had 3.1, 2.2, 2.0, 3.6, and 2.7 times more chances of carrying Gram-positive bacteria than their counterparts, respectively (Table 5).

**Table 2. Antimicrobial susceptibility patterns of Gram-positive bacteria isolated from nasopharyngeal samples in outpatient children at the UoGCSH, Northwest Ethiopia, from May 1, 2023, to August 30, 2023.**

| Bacterial isolates | Susceptibility Pattern | Antimicrobials | | | | | | | | | | | | | | |
|---|---|---|---|---|---|---|---|---|---|---|---|---|---|---|---|---|
| | | AMP N (%) | AZM N (%) | PEN N (%) | CIP N (%) | CHL N (%) | CRO N (%) | DA N (%) | DOX N (%) | ERY N (%) | FOX N (%) | CN N (%) | SXT N (%) | TET N (%) | VA N (%) | RMP N (%) |
| *S. aureus* (n = 122) | S | NT | 72 (59.0) | 2 (1.6) | 58 (47.6) | 75 (61.5) | NT | 98 (80.3) | 51 (41.8) | 85 (69.7) | 103 (84.4) | 66 (54.1) | 51 (41.8) | 28 (22.9) | NT | 102 (83.6) |
| | I | NT | - | - | 7 (5.7) | 5 (4.1) | NT | - | 6 (4.9) | - | - | 5 (4.1) | 5 (4.1) | 4 (3.3) | NT | 8 (6.6) |
| | R | NT | 50 (41.0) | 120 (98.4) | 57 (46.7) | 42 (34.4) | NT | 24 (19.7) | 65 (53.3) | 37 (30.3) | 19 (15.6) | 51 (41.8) | 66 (54.1) | 90 (73.8) | NT | 12 (9.8) |
| CoNS (n = 60) | S | NT | 37 (61.7) | 15 (25) | 38 (63.3) | 37 (61.7) | NT | 46 (78.3) | 32 (53.3) | 35 (58.3) | 57 (95) | 36 (60) | 23 (38.3) | 15 (25) | NT | 58 (96.7) |
| | I | NT | 4 (6.6) | - | - | 8 (13.3) | NT | 5 (8.3) | - | 7 (11.7) | - | 3 (5) | 6 (10) | 5 (8.3) | NT | 2 (3.3) |
| | R | NT | 19 (31.7) | 45 (75) | 22 (36.7) | 15 (25) | NT | 8 (13.3) | 28 (46.7) | 18 (30) | 3 (5) | 21 (35) | 31 (51.7) | 40 (66.7) | NT | - |
| *S. pneumoniae* (n = 92) | S | NT | NT | 60 (65.2) | NT | 62 (67.4) | NT | 78 (84.8) | 53 (57.7) | 61 (66.3) | NT | NT | 50 (54.4) | 49 (53.3) | 90 (97.8) | 87 (94.6) |
| | I | NT | NT | - | NT | - | NT | - | 4 (4.3) | - | NT | NT | 4 (4.3) | 3 (3.2) | 2 (2.2) | - |
| | R | NT | NT | 32 (34.8) | NT | 30 (32.6) | NT | 14 (15.2) | 35 (38.0) | 31 (33.7) | NT | NT | 38 (41.3) | 40 (43.5) | - | 5 (5.4) |
| *S. pyogenes* (n = 22) | S | 22 (100) | 16 (72.7) | 22 (100) | NT | 19 (86.4) | 19 (86.4) | 20 (90.9) | NT | 16 (72.7) | NT | NT | NT | 9 (40.9) | 20 (90.9) | NT |
| | I | - | 2 (9.1) | - | NT | 1 (4.5) | - | - | NT | - | NT | NT | NT | 3 (13.6) | - | NT |
| | R | - | 4 (18.2) | - | NT | 2 (9.1) | 3 (13.6) | 2 (9.1) | NT | 6 (27.3) | NT | NT | NT | 10 (45.5) | 2 (9.1) | NT |
| Total (n = 296) | S | 22 (100) | 125 (61.3) | 99 (33.4) | 87 (47.8) | 177 (59.8) | 19 (86.4) | 242 (81.8) | 136 (49.6) | 197 (66.5) | 103 (84.4) | 102 (56.0) | 124 (45.2) | 101 (34.1) | 110 (96.5) | 247 (90.1) |
| | I | - | 6 (2.9) | - | 7 (3.8) | 13 (4.4) | - | 5 (1.7) | 10 (3.7) | 7 (2.4) | - | 8 (4.4) | 15 (5.5) | 15 (5.1) | 2 (1.8) | 10 (3.7) |
| | R | - | 73 (35.8) | 197 (66.6) | 88 (48.4) | 106 (35.8) | 3 (13.6) | 49 (16.5) | 128 (46.7) | 92 (31.1) | 19 (15.6) | 72 (39.6) | 135 (49.3) | 180 (60.8) | 2 (1.8) | 17 (6.2) |

Key: S = Sensitive, I = Intermediate, R = Resistance, NT = not tested, CoNS = coagulase-negative staphylococcus species., AMP = ampicillin; AZM = azithromycin; PEN = penicillin; CIP = ciprofloxacin; CHL = chloramphenicol; CRO = ceftriaxone; DA = clindamycin; DOX = doxycycline; ERY = erythromycin; FOX = cefoxitin; CN = gentamicin; SXT = trimethoprim-sulfamethoxazole; TET = tetracycline; VA = vancomycin; RMP = rifampin

## Discussion

In the present study, the overall nasopharyngeal carriage of Gram-positive bacteria was 69.8% (95% CI: 65.3–74.0%), which is high as compared to studies of Ethiopia-Debre Berhan [2] and Turkey [17], where the overall carriage rate was 35.7% and 50%, respectively. However, our result is lower than the results reported in Ethiopia-Adama [21], Hungary [18], Sweden [39], China [40], and Nepal [41], with a carriage rate of 75.1%, 76.2%, 79.9%, 79.6%, and 94.5%, respectively. This discrepancy might be due to differences in geographical location, study populations, bacteria of interest, sample type, and sample size.

In this study, the most frequently identified isolate was *S. aureus*, 28.8% (95% CI: 24.6–33.3%). This result is consistent with results from Ethiopia-Addis Ababa (27.3%) [15]. However, the present study is higher than the studies conducted in Tanzania (23.5%) [34], Ghana (23.2%) [30], Indonesia (7.3%) [42], Nepal (16.6%) [43], and Turkey (7.9%) [17]. The reason

**Table 3. Antimicrobials susceptibility patterns of Gram-positive bacteria against erythromycin and clindamycin from nasopharyngeal samples in outpatient children at the UoGCSH, Northwest Ethiopia, from May 1, 2023, to August 30, 2023.**

| Bacterial isolates | DA$^S$, ERY$^S$ (no resistance), n (%) | DA$^S$, ERY$^R$ (D-test positive), n (%) | DA$^S$, ERY$^R$ (D-test negative), n (%) | DA$^R$, ERY$^R$ n (%) |
|---|---|---|---|---|
| *S. aureus* (n = 122) | 85 (69.7) | 10 (8.2) | 3 (2.4) | 24 (19.7) |
| MRSA (n = 19) | 6 (31.6) | 2 (10.5) | 2 (10.5) | 9 (47.4) |
| MSSA (n = 103) | 79 (76.7) | 8 (7.8) | 1 (1) | 15 (14.6%) |
| CoNS (n = 53) | 35 (66.0) | 4 (7.5) | 6 (11.3) | 8 (15.1) |
| *S. pneumoniae* (n = 92) | 61 (66.3) | 0 (0) | 17 (18.5) | 14 (15.2) |
| *S. pyogenes* (n = 22) | 16 (72.7) | 0 (0) | 4 (18.2) | 2 (9.1) |

Key. ERY$^S$: erythromycin sensitive, DA$^S$: clindamycin sensitive, ERY$^R$: erythromycin-resistant, DA$^R$: clindamycin resistant, MSSA: methicillin-sensitive *S. aureus*, MRSA: methicillin-resistant *S. aureus*, CoNS: coagulase-negative staphylococcus species.

for the high carriage in this study may be variation in the age of study participants (most of them were ≤ 3 years old in our study), and there is a possibility of acquiring *S. aureus* from parents to children through frequent close contact [44]. Moreover, these young children do not maintain their hygiene and are not mature enough in intelligence about potential pathogenic sources and disease transmission [44]. Contrarily, this study is lower than other studies in Ethiopia in Debre Berhan (53.3%) [2], Ghana (34.9%) [20], and Nepal (60.9%) [41]. Indeed, the nose of children is the preferred site to recover *S. aureus* [45], but the fact that our sample type was a nasopharyngeal swab may have led to lower *S. aureus* carriage in our study.

In this study, the prevalence of nasopharyngeal carriage was 15.6% (95% CI: 10.1–23.3) for MRSA and 5% (95% CI: 1.5–14.8) for MRCoNS. This is concordant with the study conducted in Iran (21.7%) [46] and Germany (7%) [47], respectively. However, this MRSA is higher than studies observed in Ethiopia-Addis Ababa (2.7%) [15], Sudan (8.5%) [48], Nigeria (5.3%) [49], and Spain (1.44%) [50]. On the other hand, our result of MRSA is lower than that of Ethiopia-Debre Berhan (79.5%) [2], Uganda (31.3%) [8], and China (42.4%) [51]. The high proportion of healthcare institution visits observed and the fact that more than half of our study participants responded to having antimicrobial use practice might partly reflect the high MRSA carriage in our investigation. Moreover, it is relevant to note that upon visiting healthcare institutions, health workers may introduce MRSA via physical contact with children [52].

In this study, the nasopharyngeal carriage of *S. pneumoniae* was 21.7% (95% CI: 18.0–25.9%). This is comparable to studies conducted in Ethiopia-Hawassa (21.5%) [53] and the Democratic Republic of the Congo (21%) [29]. Nevertheless, it is higher than earlier studies reported from Ethiopia-Bahir Dar (10.3%) [54], Thailand (12.9%) [55], and China (5.5%) [56]. The high carriage of this bacterium may result from inadequate coverage of vaccination, the spread of pneumococci among individuals, and booster dose insufficiency at the age of two years, which subsequently weaken immunity against specific serotypes [57]. On the other hand, this study is lower than other studies in Ethiopia-Jimma (43.8%) [31], Uganda (58.6%) [58], Nepal (42.9%) [41], and Indonesia (45%) [26]. The inclusion of children aged above five years in our study may have resulted in low results compared to the studies mentioned above, which target only children aged under five years. This is strengthened by the fact that the *S. pneumoniae* carriage rate increases mostly in under-five children and decreases afterward [18].

In the present study, the nasopharyngeal carriage for *S. pyogenes* was 5.2% (95% CI: 3.4–7.8%). This is similar to the studies conducted in Ethiopia-Debre Berhan (7.3%) [2] and Nepal (5.3%) [43]. Nevertheless, it is higher than the study in Turkey (2.9%) [17]. On the other hand, it is lower than earlier studies in Ethiopia-Jigjiga (10.6%) [6], Uganda (15.9%) [59], and Yemen (12.8%) [60]. The reason for the difference in carriage rate might be partly related to variations

**Table 4. Multi-drug resistance profile of Gram-positive bacteria isolated from nasopharyngeal samples in outpatient children at UoGCSH, Northwest Ethiopia, from May 1, 2023, to August 30, 2023.**

| Resistance patterns | Antimicrobials (classes) | Gram-positive bacterial isolates | | | | |
|---|---|---|---|---|---|---|
| | | *S. aureus* (n = 122) | *S. pneumoniae* (n = 92) | *S. pyogenes* (n = 22) | CoNS (n = 60) | Total (n = 296) |
| Susceptible for the tested antimicrobials | - | 2 | 12 | 11 | 14 | 39 |
| PEN | 1 (1) | 9 | 7 | - | 2 | 18 |
| TET | 1 (1) | 5 | 13 | 5 | 4 | 27 |
| SXT | 1 (1) | 3 | 2 | - | 6 | 11 |
| ERY, AZM | 2 (1) | 1 | - | 1 | - | 2 |
| TET, DOX | 2 (1) | 2 | 3 | - | - | 5 |
| PEN, TET | 2 (2) | 2 | 2 | - | 3 | 7 |
| PEN, TET, DOX | 3 (2) | 1 | 2 | - | - | 3 |
| PEN, SXT | 2 (2) | 2 | 1 | - | 1 | 4 |
| PEN, CHL | 2 (2) | 1 | 1 | - | - | 2 |
| PEN, DA | 2 (2) | 1 | 1 | - | 1 | 3 |
| PEN, ERY | 2 (2) | 1 | 1 | - | 1 | 3 |
| PEN, ERY, AZM | 3 (2) | 1 | - | - | - | 1 |
| TET, SXT | 2 (2) | 2 | 4 | - | 2 | 8 |
| TET, CHL | 2 (2) | 1 | 2 | - | 1 | 4 |
| TET, ERY | 2 (2) | 1 | 1 | - | 1 | 3 |
| TET, DA | 2 (2) | 1 | 1 | - | - | 2 |
| SXT, CHL | 2 (2) | 1 | 2 | - | - | 3 |
| CHL, ERY | 2 (2) | 1 | 1 | - | - | 2 |
| ERY, CHL, AZM | 3 (2) | 1 | - | - | - | 1 |
| ERY, DA | 2 (2) | 1 | - | - | - | 1 |
| TET, AZM, ERY | 3 (2) | - | - | 1 | - | 1 |
| ERY, DA, CHL | 3 (3) | 1 | 1 | 1 | - | 3 |
| PEN, TET, ERY | 3 (3) | 4 | 1 | - | 2 | 7 |
| TET, ERY, DA | 3 (3) | 3 | 2 | 1 | - | 6 |
| SXT, ERY, DA | 3 (3) | 2 | 1 | - | - | 3 |
| PEN, TET, CHL, DOX | 4 (3) | 3 | 1 | - | - | 4 |
| PEN, TET, SXT, DOX | 4 (3) | 7 | 1 | - | 2 | 10 |
| PEN, TET, DA | 3 (3) | 3 | 1 | - | 2 | 6 |
| PEN, DA, CIP | 3 (3) | 1 | - | - | 2 | 3 |
| PEN, ERY, DA | 3 (3) | 2 | 1 | - | 1 | 4 |
| TET, SXT, CHL, DOX | 4 (3) | 4 | 1 | - | 2 | 7 |
| TET, ERY, CHL, DOX | 4 (3) | 1 | 2 | - | 1 | 4 |
| PEN, TET, SXT, CIP | 4 (4) | 2 | - | - | 2 | 4 |
| PEN, TET, SXT, AZM, DOX | 5 (4) | 4 | - | - | 1 | 5 |
| PEN, TET, SXT, ERY, DOX | 5 (4) | 6 | 4 | - | 1 | 11 |
| PEN, TET, SXT, CHL, DOX | 5 (4) | 10 | 8 | - | 1 | 19 |
| PEN, TET, SXT, DA | 4 (4) | 3 | 2 | - | 1 | 6 |
| TET, ERY, AZM, CRO | 4 (4) | - | - | 2 | - | 2 |
| PEN, TET, SXT, ERY, AZM, DA | 5 (5) | 1 | - | - | - | 1 |
| PEN, TET, SXT, ERY, CIP | 5 (5) | 1 | - | - | 1 | 2 |
| PEN, TET, SXT, ERY, CN | 5 (5) | 1 | - | - | 2 | 3 |
| PEN, TET, SXT, ERY, CHL | 5 (5) | 3 | 4 | - | 1 | 8 |
| PEN, TET, SXT, ERY, DA | 5 (5) | 2 | 1 | - | - | 3 |
| PEN, TET, SXT, ERY, DA, DOX | 6 (5) | 3 | 1 | - | - | 4 |

*(Continued)*

**Table 4.** (Continued)

| Resistance patterns | Antimicrobials (classes) | Gram-positive bacterial isolates | | | | |
|---|---|---|---|---|---|---|
| | | S. aureus (n = 122) | S. pneumoniae (n = 92) | S. pyogenes (n = 22) | CoNS (n = 60) | Total (n = 296) |
| PEN, TET, SXT, ERY, DA, CHL | 6 (6) | 5 | 2 | - | 1 | 8 |
| PEN, TET, SXT, CHL, AZM, CN | 6 (6) | 2 | - | - | 1 | 3 |
| PEN, TET, SXT, CHL, ERY, CIP | 6 (6) | 1 | - | - | - | 1 |
| PEN, TET, SXT, CHL, ERY, DA, DOX | 7 (6) | 4 | 2 | - | - | 6 |
| PEN, TET, SXT, CHL, AZM, DOX, CIP | 7 (6) | 1 | - | - | - | 1 |
| PEN, TET, SXT, CHL, DOX, AZM, ERY, DA | 8 (6) | 2 | - | - | - | 2 |
| Total | MDR | 82 (67.2%) | 36 (39.1%) | 4 (18.2%) | 24 (40%) | 146 (49.3%) |
| | Non-MDR | 40 (32.8%) | 56 (60.9%) | 18 (81.8%) | 36 (60%) | 150 (50.7%) |

Key: AZM = azithromycin; PEN = penicillin; CIP = ciprofloxacin; CHL = chloramphenicol; CRO = ceftriaxone; DA = clindamycin; DOX = doxycycline; ERY = erythromycin; CN = gentamicin; SXT = trimethoprim-sulfamethoxazole; TET = tetracycline; CoNS = Coagulase-negative staphylococcus species; MDR = multi-drug resistant (to $\geq$ 3 antimicrobial classes)

that existed in circulating strains of *S. pyogenes* and the immunity of study participants against these strains [61]. Additionally, earlier studies used a throat swab, whereas this study used a nasopharyngeal swab, leading to a difference in carriage rate.

In this study, high resistance by *S. aureus* isolates against penicillin (98.4%) was found. Some comparable findings from earlier literature were documented in Ghana [20] and Hungary [18], where 84.5% and 73.7–82.8% resistance have been registered, respectively. This resistance may be because of beta-lactamase production and alteration of the penicillin-binding proteins by *S. aureus* [2].

In the present study, we found that *S. pneumoniae* isolates demonstrated greater resistance to trimethoprim-sulfamethoxazole (41.3%), tetracycline (43.5%), and oxacillin-1µg (penicillin) (34.8%) than the other tested antimicrobials. This study corresponds with a previous study in Ethiopia-Jimma [31], where resistance to tetracycline (43.7%), trimethoprim-sulfamethoxazole (38.0%), and penicillin (36.1%) have been documented. However, this study is lower than the studies in Ethiopia-Hawassa (48.5%) [53], Morocco (57.2%) [24], Ethiopia-Debre Berhan (68.8%) for tetracycline [3] and Ethiopia-Hawassa (64.6%) [53] for trimethoprim-sulfamethoxazole. Moreover, differences in the techniques employed to measure AST, antimicrobial consumption, isolate source, and geographical location might have benefits for this variation [62].

In this study, erythromycin resistance was (33.7%) and (30.3%%) to *S. pneumoniae* and *S. aureus*, respectively, and inducible clindamycin resistance was 8.2% to *S. aureus and* 0 (0%) to *S. pneumoniae*. This result is comparable with the studies in Ethiopia-Arba Minch (27.8%) [63] and Sudan (30.5%) [48] for *S. aureus*. However, it is higher than that of the study in Ethiopia-Jimma [31] and Malaysia [64], with erythromycin resistance of 5.7% and 0% for *S. pneumoniae* and *S. aureus*, respectively. In contrast, our finding contradicts the previous studies conducted in Morocco [24] and Tanzania [65] that present inducible clindamycin resistance of 4.5% and 16.7% for *S. pneumoniae* and *S. aureus*, respectively. The variation observed can be ascribed to the different methods taken in interpreting the findings; EUCAST was utilized in the previous study, whereas CLSI was used in this study. In addition, differences in geographic regions, institutional settings, strain types, and antimicrobial prescription approaches may all have an impact on the occurrence of variations in inducible clindamycin resistance [64].

The MDR proportions of 69.2%, 40%, 39.1%, and 18.2% were found in the current study for *S. aureus*, CoNS, *S. pneumoniae*, and *S. pyogenes*, respectively. This MDR is comparable

**Table 5. Bivariable and multivariable logistic regression analysis of factors associated with nasopharyngeal carriage of Gram-positive bacteria among children attending the outpatient department at UoGCSH, Northwest Ethiopia, from May 1, 2023, to August 30, 2023.**

| Characteristics of study participants (N = 424) | | Nasopharyngeal carriage | | COR (95% CI) | P-value | AOR (95% CI) | P-value |
|---|---|---|---|---|---|---|---|
| | | Yes, N (%) | No, N (%) | | | | |
| Sex | Male | 163 (71.5) | 65 (28.5) | 1.188 (0.784–1.800) | 0.417 | - | - |
| | Female | 133 (67.9) | 63 (32.1) | 1 (Ref.) | | | |
| Age | <3 years | 108 (72) | 42 (28) | 0.896 (0.489–1.642) | 0.723 | 1.120 (0.545–2.301) | 0.757 |
| | 3–5 years | 63 (60.6) | 41 (39.4) | 1.500 (0.800–2.810) | 0.206 | 1.754 (0.843–3.648) | 0.132 |
| | 6–10 years | 72 (76.6) | 22 (23.4) | 0.704 (0.355–1.395) | 0.315 | 0.852 (0.388–1.871) | 0.690 |
| | 11–15 years | 53 (69.7) | 23 (30.3) | 1 | | 1 | |
| Residence | Urban | 174 (69.3) | 77 (30.7) | 1 | | | |
| | Rural | 122 (70.5) | 51 (29.5) | 0.945 (0.619–1.442) | 0.792 | - | - |
| Mothers' education | Illiterate | 38 (53.5) | 33 (46.5) | 2.437 (1.310–4.535) | 0.005 | 3.639 (1.691–7.829) | **0.001**\*\* |
| | Elementary completed | 75 (68.2) | 35 (31.8) | 1.310 (0.738–2.324) | 0.357 | 1.498 (0.763–2.941) | 0.241 |
| | Secondary completed | 96 (76.8) | 29 (23.2) | 0.848 (0.473–1.520) | 0.579 | 1.243 (0.615–2.510) | 0.545 |
| | Diploma and above | 87 (73.7) | 31 (26.3) | 1 | | 1 | |
| Fathers' education | Illiterate | 55 (64.7) | 30 (35.3) | 1.144 (0.617–2.120) | 0.670 | - | - |
| | Elementary completed | 87 (72.5) | 33 (27.5) | 0.795 (0.443–1.429) | 0.444 | - | - |
| | Secondary completed | 89 (72.4) | 34 (27.6) | 0.801 (0.447–1.434) | 0.455 | - | - |
| | Diploma and above | 65 (67.7) | 31 (32.3) | 1 | | | |
| Fathers' occupation | Governmental | 116 (72.0) | 45 (28.0) | 1 | | 1 | |
| | Merchant | 102 (72.3) | 39 (27.7) | 0.986 (0.595–1.633) | 0.955 | 0.897 (0.493–1.630) | 0.720 |
| | Farmers | 29 (64.4) | 16 (35.6) | 1.422 (0.706–2.866) | 0.325 | 1.322 (0.500–3.500) | 0.574 |
| | Daily labour | 49 (63.6) | 28 (36.4) | 1.473 (0.826–2.626) | 0.189 | 1.375 (0.632–2.993) | 0.422 |
| Mothers' occupation | Governmental | 106 (74.6) | 36 (25.4) | 1 | | 1 | |
| | Merchant | 78 (69.6) | 34 (30.4) | 1.283 (0.739–2.230) | 0.376 | 0.913 (0.427–1.949) | 0.813 |
| | Housewife | 74 (67.9) | 35 (32.1) | 1.393 (0.802–2.418) | 0.240 | 1.163 (0.603–2.243) | 0.653 |
| | Daily labour | 38 (62.3) | 23 (37.7) | 1.782 (0.939–3.384) | 0.077 | 1.449 (0.588–3.572) | 0.420 |
| Income | < 500 | 68 (85.0) | 12 (15.0) | 0.373 (0.183–0.757) | 0.006 | 0.546 (0.233–1.277) | 0.163 |
| | 501–1000 | 57 (65.5) | 30 (34.5) | 1.111 (0.630–1.958) | 0.716 | 1.367 (0.685–2.729) | 0.375 |
| | 1001–1500 | 76 (65.0) | 41 (35.0) | 1.139 (0.677–1.915) | 0.624 | 0.982 (0.512–1.885) | 0.957 |
| | >1500 | 95 (68.0) | 45 (32.0) | 1 | | 1 | |
| Family size | <5 | 150 (81.5) | 34 (18.5) | 1 | | 1 | |
| | ≥5 | 146 (60.8) | 94 (39.2) | 2.840 (1.805–4.471) | 0.000 | 3.061 (1.595–5.874) | **0.001**\*\* |
| The habit of frequent nose-picking | Yes | 148 (69.8) | 64 (30.2) | 1.000 (0.661–1.514) | 1.000 | - | - |
| | No | 148 (69.8) | 64 (30.2) | 1 | | | |
| Bed-sharing with parents | Yes | 114 (67.5) | 55 (32.5) | 1.203 (0.790–1.833) | 0.390 | - | - |
| | No | 182 (71.4) | 73 (28.6) | 1 | | | |
| Siblings < 5 years | Yes | 142 (64.5) | 78 (35.5) | 1.692 (1.109–2.580) | 0.015 | 1.991 (1.196–3.313) | **0.008**\*\* |
| | No | 154 (75.5) | 50 (24.5) | 1 | | 1 | |
| Siblings ≥ 5 years | Yes | 145 (70.7) | 60 (29.3) | 0.919 (0.607–1.392) | 0.690 | - | - |
| | No | 151 (68.9) | 68 (31.1) | 1 | | | |
| Kitchen location | Indoor | 139 (61.8) | 86 (38.2) | 2.313 (1.499–3.569) | 0.000 | 2.195 (1.275–3.778) | **0.005**\*\* |
| | Outdoor | 157 (78.9) | 42 (21.1) | 1 | | 1 | |
| Smoker in house | Yes | 19 (63.3) | 11 (36.7) | 1.371 (0.633–2.970) | 0.424 | - | - |
| | No | 277 (70.3) | 117 (29.7) | 1 | | 1 | |
| Number of rooms per house | < 2 | 135 (66.5) | 68 (33.5) | 1.871 (1.065–3.286) | 0.029 | 1.138 (0.525–2.464) | 0.743 |
| | 3–4 | 83 (68.0) | 39 (32.0) | 1.745 (0.945–3.225) | 0.075 | 1.090 (0.469–2.536) | 0.841 |
| | ≥ 5 | 78 (78.8) | 21 (21.2) | 1 | | 1 | |

*(Continued)*

**Table 5.** (Continued)

| Characteristics of study participants (N = 424) | | Nasopharyngeal carriage | | COR (95% CI) | P-value | AOR (95% CI) | P-value |
|---|---|---|---|---|---|---|---|
| | | Yes, N (%) | No, N (%) | | | | |
| Previous respiratory tract infection | Yes | 136 (65.1) | 73 (34.9) | 1.561(1.028–2.372) | 0.037 | 1.198 (0.729–1.968) | 0.476 |
| | No | 160 (74.4) | 55 (25.6) | 1 | | | |
| Sign of runny nose for the last two weeks | Yes | 198 (71.2) | 80 (28.8) | 0.825 (0.536–1.271) | 0.383 | - | - |
| | No | 98 (67.1) | 48 (32.9) | 1 | | | |
| Vaccination status | Not vaccinated | 11 (57.9) | 8 (42.1) | 1.561 (0.607–4.015) | 0.356 | 2.302 (0.714–7.424) | 0.163 |
| | Partially vaccinated | 94 (75.2) | 31 (24.8) | 0.708 (0.439–1.141) | 0.156 | 0.785 (0.430–1.436) | 0.432 |
| | Fully vaccinated | 191 (68.2) | 89 (31.8) | 1 | | 1 | |
| Recent hospitalisation | Yes | 42 (56.8) | 32 (43.2) | 2.016 (1.203–3.379) | 0.08 | 1.763 (0.945–3.289) | 0.075 |
| | No | 254 (72.6) | 96 (27.4) | 1 | | 1 | |
| Receive recent antimicrobials | Yes | 153 (70.5) | 64 (29.5) | 0.935 (0.617–1.415) | 0.749 | - | - |
| | No | 143 (69.1) | 64 (30.9) | 1 | | | |
| History of visiting healthcare institutions | Yes | 228 (67.9) | 108 (32.1) | 1.611 (0.930–2.788) | 0.089 | 2.690 (1.405–5.151) | **0.003**** |
| | No | 68 (77.3) | 20 (22.7) | 1 | | 1 | |

Key: AOR = adjusted odds ratio, CI = confidence interval, COR = crude odds ratio, ** = Significant at P-value < 0.05

with the studies in Ethiopia (Debre Berhan) and Indonesia, which found *S. aureus* (68.5%) [2] and *S. pneumoniae* (36%) [66], respectively. However, a higher MDR result was observed in our study as compared to studies in Nepal (51.3%) [41] and Ghana (3.2%) [30] for *S. aureus*, in Germany (23%) for CoNS [47], and in Ethiopia (17.7%, 2.9%) [31, 53] and Indonesia (18%) [26] for *S. pneumoniae* isolates. The occurrence of high MDR could be because bacteria produce genetic changes (*S. aureus*, for example, alter DNA gyrase or minimize outer membrane proteins), reduce the uptake of antimicrobials, change the target site, and get out of the site of action, leading to antimicrobial resistance [67].

The reason for the rise in MDR in *S. pneumoniae* in this study could be partly related to the vaccine effects, antimicrobial consumption, and the production and dissemination of antimicrobial-resistant strains [68]. However, our MDR results for *S. pneumoniae* and *S. pyogenes* were lower than the findings of a recently published article in Ethiopia (Debre Berhan), with an MDR of 68.8% and 40%, respectively [3].

The current study presented that children living with sibling(s) < 5 years old had a greater likelihood of nasopharyngeal Gram-positive bacteria carriage when compared to children who were not living with younger sibling(s) < 5 years old (AOR = 2.001, 95% CI: 1.207–3.319, P = 0.007). This association has also been documented in earlier studies in Ethiopia and Italy [31, 53, 69]. The association between nasopharyngeal bacteria carriage and the existence of siblings < 5 years is explicable, as younger sibling(s) < 5 years usually have a less developed immune system and contain high bacterial density in their nasopharynx [31]. They are, therefore, efficient at transmitting bacteria to other healthy children and the general population as well, horizontally.

This study revealed that nasopharyngeal Gram-positive bacteria carriage is significantly more likely in cases where the mother is illiterate (AOR = 3.675, 95% CI (1.718–7.860, P = 0.001). In the same manner, Ethiopian (P = 0.0012) [3] and Brazilian studies (P = 0.022) [70] have also reported a noteworthy relationship between having illiterate parents and nasopharyngeal bacteria carriage. This makes sense because illiterate mothers had a poorer understanding of hygienic conditions such as personal hygiene, handwashing practices, and not using materials in common than literate parents [71].

In our study, it was found that children from households with more than five members had higher odds of carrying Gram-positive bacteria in the nasopharynx compared to those from smaller families (AOR = 3.168, 95% CI: 1.620–6.196, P = 0.001). Similar associations between large family size and nasopharyngeal bacterial carriage have been observed in studies conducted in Ethiopia, specifically in Debre Berhan (P = 0.006) [2], and in China (P = 0.043) [72]. This link may be explained by the overcrowding typical of larger families, which can lead to reduced ventilation, increased strain variation through horizontal gene transfer, and enhanced aerosol transmission of bacteria among family members, including from carriers to non-carriers [2, 73].

This study presented that indoor cooking emerged as another significant risk factor associated with nasopharyngeal carriage of Gram-positive bacteria (AOR = 2.182, 95% CI: 1.274–3.736, P = 0.004). Similar findings have been reported in studies conducted in Ethiopia-Hawassa (P = 0.002) [53], and in Kenya [74] have also demonstrated a comparable finding to our study. The potential reason is that indoor cooking releases air pollutants that can irritate the respiratory tract, diminish immune function by inducing inflammation and oxidative stress, impair immune cell responses, and reduce blood oxygen transportation. These factors collectively contribute to increased bacterial carriage among household members [75–77].

## Limitations of the study

Even though this study provides pertinent information about the nasopharyngeal carriage, antimicrobial susceptibility patterns, and associated factors of Gram-positive bacteria among children attending the outpatient department at UoGCSH, the minimum inhibitory concentration for penicillin-resistant *S. pneumoniae* and vancomycin-resistant *S. aureus* and species identification of CoNS were not determined due to resource scarcity.

## Conclusions and recommendations

The study revealed an overall nasopharyngeal carriage of 69.8% in Gram-positive bacteria among outpatient children, with *S. aureus* being the most commonly detected bacterium. Many of the isolates exhibited significant resistance to penicillin, tetracycline, and trimethoprim-sulfamethoxazole, with *S. aureus* emerging as the predominant MDR strain and notable MRSA, while *S. pyogenes* showed higher susceptibility. Clindamycin, rifampin, and erythromycin demonstrated the highest efficacy among the antimicrobials tested. Associated factors for bacterial carriage included healthcare visits, larger family sizes, younger siblings, maternal illiteracy, and indoor cooking. The study emphasizes the importance of monitoring MRSA carriage in pediatric outpatient settings. Community health education by the Regional Health Bureau is necessary, particularly for the guardians of children. Additionally, families should be encouraged to have separate kitchens from sleeping areas to improve ventilation and reduce bacterial transmission. Regular screening of younger siblings in healthcare settings is also recommended to control the spread of bacteria. Moreover, the study called for hospital and community-based studies with advanced techniques such as minimum inhibitory concentration testing and molecular characterization to better understand the resistance patterns as well as resistance genes of circulating bacteria.

## Acknowledgments

The authors are very grateful to the Department of Medical Microbiology, School of Biomedical and Laboratory Sciences, College of Medicine and Health Sciences, University of Gondar. We would also like to give our sincere thanks to the staff members working in the outpatient

pediatrics department at the University of Gondar Comprehensive Specialized Hospital. Lastly, we would like to acknowledge the study participants.

## Author Contributions

**Conceptualization:** Abebe Birhanu, Azanaw Amare, Mitkie Tigabie, Feleke Moges.

**Data curation:** Abebe Birhanu, Eden Getaneh, Tena Cherkos.

**Formal analysis:** Abebe Birhanu, Azanaw Amare, Mitkie Tigabie, Eden Getaneh, Tena Cherkos, Feleke Moges.

**Funding acquisition:** Abebe Birhanu.

**Investigation:** Abebe Birhanu.

**Methodology:** Abebe Birhanu, Azanaw Amare, Mitkie Tigabie, Muluneh Assefa, Feleke Moges.

**Project administration:** Abebe Birhanu.

**Resources:** Abebe Birhanu.

**Software:** Abebe Birhanu, Mitkie Tigabie, Eden Getaneh, Tena Cherkos.

**Supervision:** Azanaw Amare, Mitkie Tigabie, Muluneh Assefa, Feleke Moges.

**Validation:** Abebe Birhanu, Azanaw Amare, Mitkie Tigabie, Feleke Moges.

**Visualization:** Abebe Birhanu, Feleke Moges.

**Writing – original draft:** Abebe Birhanu, Azanaw Amare, Mitkie Tigabie, Eden Getaneh, Muluneh Assefa, Tena Cherkos, Feleke Moges.

**Writing – review & editing:** Abebe Birhanu, Azanaw Amare, Mitkie Tigabie, Eden Getaneh, Muluneh Assefa, Tena Cherkos, Feleke Moges.

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
