## [Decision Letter · Decision Letter 0]

1 May 2024

PONE-D-24-10619Nasopharyngeal Carriage, Antibiotic Susceptibility Patterns, and Associated Factors of Gram-positive Bacteria among children attending the outpatient department at the University of Gondar Comprehensive Specialized Hospital, Northwest EthiopiaPLOS ONE

Dear Dr. Birhanu,

Thank you for submitting your manuscript to PLOS ONE. After careful consideration, we feel that it has merit but does not fully meet PLOS ONE’s publication criteria as it currently stands. Therefore, we invite you to submit a revised version of the manuscript that addresses the points raised during the review process.

We look forward to receiving your revised manuscript.

Kind regards,

Tebelay Dilnessa, MSc

Academic Editor

PLOS ONE

Journal Requirements:

2. Thank you for submitting the above manuscript to PLOS ONE. During our internal evaluation of the manuscript, we found significant text overlap between your submission and previous work in the [introduction, conclusion, etc.].

Please revise the manuscript to rephrase the duplicated text, cite your sources, and provide details as to how the current manuscript advances on previous work. Please note that further consideration is dependent on the submission of a manuscript that addresses these concerns about the overlap in text with published work.

[If the overlap is with the authors’ own works: Moreover, upon submission, authors must confirm that the manuscript, or any related manuscript, is not currently under consideration or accepted elsewhere. If related work has been submitted to PLOS ONE or elsewhere, authors must include a copy with the submitted article. Reviewers will be asked to comment on the overlap between related submissions (http://journals.plos.org/plosone/s/submission-guidelines#loc-related-manuscripts).]

We will carefully review your manuscript upon resubmission and further consideration of the manuscript is dependent on the text overlap being addressed in full. Please ensure that your revision is thorough as failure to address the concerns to our satisfaction may result in your submission not being considered further.

"The authors' have no conflict of interest"

Additional Editor Comments:

Although what authors are reporting is not a novelty, they must however be commended for their efforts in reporting what is prevailing in their localityThe email addresses of authors were not needed here in the main manuscript as it appears in the system except the corresponding author.Based on PLoS One authors’ guideline, figures must be uploaded separately in tif or other form.
Line 27: ‘…was conducted from May 1 to August 30, 2023.’ It is better written as, ‘…was conducted from May 1, 2023 to August 30, 2023.’Lines 29&30: ‘Bacterial species were identified by biochemical tests.’ It is rewritten better as, ‘Bacterial species were identified by colony morphology, Gram stain and biochemical tests.’Lines 31 &32: ‘…… were determined using the cefoxitin disc and D-test, respectively.’ Do you think ‘cefoxitin disc and D-test’ are equivalent terms?  Drug Vs technique?Lines 46: Write their long form of the ‘MRSA and MDR’The abstract should be re**-**written and make adjustments based on the comments in the methods and results part that follows.
Once you have used the long form of the name of the bacterium then you can abbreviate the genus name throughout the body of the paper, except headings and subheadings.The objective was not needed. Therefore, remove the objective (lines from 104- 115).
Line 123: The total population of the town is estimated to be 412,739 by the year 2023 [28]. Here this is misleading as the population of Gondar town is not only 412,739 by 2023. It is expected above 1million. Please revise it.A separate subheading is needed as, ‘Study population’A separate subheading is needed as, ‘Sample size determination and sampling technique’Variables of the study should be removed (lines 140-152).Why you missed to include chocolate agar especially for isolation of fastidious organisms like *S. peumoniae?*What is the advantage of doing oxacillin and cefoxitin disc diffusion test concurrently? Can we perform oxacillin susceptibility test by disc diffusion for *S. aureus*?Why you miss to do ‘vancomycin susceptibility testing’, the best drug of choice for MRSA?
During presenting the result some requires both the numerator and denominator. Similarly, associated factor assessment date requires P- value.All table titles should be complete in description, area and period of study.
Most of the reason for variation mentioned in the discussion part were already intensively described by other authors.
The conclusion was beyond your scope so revise it based on community/government importance (try to select the most important one).Line 479-484: Authors contribution should be removed as the system creates automatically.It is better also you follow the manuscript writing protocol for PloS One, especially font size, font type, reference list writing, table and figure preparation and whether figures submitted within the main manuscript or not. Additionally, binomial nomenclature for name of the bacteria was not appropriately followed.

Reviewers' comments:

Reviewer's Responses to Questions

**Comments to the Author**

1. Is the manuscript technically sound, and do the data support the conclusions?

Reviewer #1: Partly

Reviewer #2: Yes

2. Has the statistical analysis been performed appropriately and rigorously? 

Reviewer #1: No

Reviewer #2: Yes

3. Have the authors made all data underlying the findings in their manuscript fully available?

Reviewer #1: Yes

Reviewer #2: Yes

4. Is the manuscript presented in an intelligible fashion and written in standard English?

Reviewer #1: Yes

Reviewer #2: No

5. Review Comments to the Author

Reviewer #1: Manuscript Number: PONE-D-24-10619

Manuscript Title: Nasopharyngeal Carriage, Antibiotic Susceptibility Patterns, and Associated Factors of Gram-positive Bacteria among children attending the outpatient department at the University of Gondar Comprehensive Specialized Hospital, Northwest Ethiopia

Dear editor,

I would like to say thanks for giving the opportunity and the invitation. It was my pleasure to review this scientific paper. I have provided comments, reconditions, and question to the authors below:

Point by Point comments to Authors

1. The main concern of this paper is it lacks novelty. There are many researches in Ethiopia in general and even at University of Gondar Comprehensive Specialized Hospital in particular on the Nasopharyngeal Carriage and Antibiotic Susceptibility Patterns of Gram-positive isolates: (DOI:10.1016/j.pedneo.2013.03.017DOI: 10.1016/j.pedneo.2013.03.017, https://doi.org/10.3390/children4040027, https://doi.org/10.1177/23333928231186687, DOI:10.4172/2332-0877.1000109, DOI: 10.2147/IDR.S385866…..). Thus, it is better to supplement some important issues like genotypic characterizations of the isolates via extended your study time, and the study groups (please include the inpatients).

2. Why you focus only on Gram-positive bacteria? For example Gram negative bacteria like Haemophilus influenzae, Klebsiella pneumoniae, Micrococcus catarrhalis ………. Can induce Nasopharyngeal Carriage

3. What about the viral and the fungal agents? As there are some fungal and viral agents, which cause Nasopharyngeal Carriage (DOI: 10.1186/s41479-021-00088-5, https://pubmed.ncbi.nlm.nih.gov/23798081/)

4. Line number 32. Please convert Epi-Data version 4.6.0.6 to 4.6

5. Line number 43: Please convert risk factors to associated factors

6. In the abstract section, minimize the conclusions and the recommendations and omit recommendation as sub-title in the abstract part.

7. In line number 50, please remove colon (:) in front of the introduction

8. The introduction section needs some modification as it focuses on the scientific background. Hence, it will be grateful if you start from the problem with respect to the WHO and CDC data.

9. Please state the problem and prevalence of nasopharyngeal carriage and drug resistance profiles of the isolates among children in Ethiopia and the study area in figures?

10. Online number 105, please convert general objectives to general objective?

11. Why you excluded under 15 ages Children and those with a history of nasal surgery, respiratory infection, and antibiotics from the study?

12. You have used a single population proportion formula to calculate the sample size. Thus, please write the formula and how to obtained 424 and why you used 50% proportion? I think there are plenty of articles even in your study area.

13. Who analyzed the data? You said a trained laboratory technologist analyzed the data (on line numbers 155 and 156).

14. On line number 238, please change p-value ≤ 0.2 to 0.25

15. Please omit the natural resistance profiles of the isolates (those which have known resistance profile to certain antibiotics). For instance, S. aureus naturally resistant to penicillin, clindamycin, and azithromycin.

16. Online number 305, please covert 257 to text as it comes before the full stop.

17. In the discussion section, please state the scientific justifications to associate factors

18. It is better if you minimize the Conclusions and Recommendations statements as it is so bulky

19. Online 461 and 462, avoid the figure (69.8%) and write as low, intermediate, and high

20. In the authors’ contributions statement, please avoid the bolded letters

21. Please re-write the competing of interest and funding statement

22. There are many too old references in the manuscript (ref 2, 3, 4, 18, 20, 25, 27, 30, 31, 34, 41, 44, 49, 54, 68, 69, 76, 78…..). Hence, please update and replace them.

Question

1. What is the clinical utility and value of your study? In your study, the most frequently identified isolate was S. aureus. However, S. aureus is the known causative agent for Nasopharyngeal Carriage. Moreover, Gram-positive bacteria, including S. aureus, Streptococcus pyogens, and Streptococcus pneumoniae are the cause of Nasopharyngeal Carriage naturally. Thus, what is the importance doing this research?

Reviewer #2: Dear Author,

I wish to acknowledge the author's work entitled: Nasopharyngeal Carriage, Antibiotic Susceptibility Patterns, and Associated Factors of Gram-positive Bacteria among children attending the outpatient department at the University of Gondar Comprehensive Specialized Hospital, Northwest Ethiopia. The results illustrated the carriage state of Gram-positive bacteria among children. Therefore, this study will be useful for the prevention and control of the carriage state of Gram-positive bacteria among children in the study area. The manuscript is more or less in line with the vision, objectives, and instructions of PLoS One. Despite its potential, however, this manuscript contains major errors in data presentation and interpretation, grammatically which need to be corrected.

General comments

1. The study mixes carriage isolates with what are likely to be clinical isolates.

2. Why do you exclude Gram-negative bacteria?

3. Uniform writing is needed for the name of drugs.

4. The use of the English language is poor in certain sections and would require a detailed revision.

Abstract

• Lines 32-34: ‘The data was entered into Epi-Data version 4.6.0.6 and exported to SPSS version 25 for analysis. The adjusted odds ratio at a 95% confidence interval with a P-value of < 0.05 was taken as statistically significant.’ It is better written as The data was entered into Epi-Data version 4.6.0.6 and exported to SPSS version 25 for analysis. Bivariable and multivariable logistic regression analysis was used to identify associated risk factors. The adjusted odds ratio at a 95% confidence interval with a P-value of < 0.05 was taken as statistically significant.’

• The number of isolates with the total is better written together with the percentages. Resistant/susceptible with the total including the percentage also should be used. This works for the whole abstract and results part of the paper. For example, Lines 35 &36: ‘The overall nasopharyngeal carriage of Gram-positive bacteria was 296/424 (69.8%, 95% CI: 65.3–74.0). Of these, S. aureus constitutes 122/424 (28.8%), followed by S. pneumoniae 92/424 (21.7%).’ It is better written as, ‘‘The overall nasopharyngeal carriage of Gram-positive bacteria was 296/424 (69.8%, 95% CI: 65.3–74.0). Of these, S. aureus constitutes 122/424 (28.8%), followed by S. pneumoniae 92/424 (21.7%).’

• Lines 40-43: P- value is needed.

• Line 43: …… risk factors……. It is better written as……. associated factors…….

Introduction

• Once you have used the long form of the name of the bacterium then you can abbreviate the genus name throughout the body of the paper, except headings and subheadings.

Materials and Methods

• Line 53: Remove the word ‘analysis’

• Specimen collection and processing needs to be well written.

• What standard microbiological procedures were employed in specimen collection, isolation, identification susceptibility testing, and throughout the procedure?

• How could you standardize the suspension preparation while performing antimicrobial susceptibility testing?

• What is the advantage of doing cefoxitin disc diffusion over oxacillin disc for susceptibility testing for methicillin (MRSA)?

• Better say, ‘Data analysis’ than ‘Statistical analysis’

• What have you done for children positive for the microorganisms?

Results

• How can you define community-associated methicillin-resistant S. aureus (CA-MRSA)? How can you exclude nosocomial sources?

• What is your base for classifying the age group? Have you associated with possible risks of interest? It is better again to re-group and perform the analysis.

Discussions

• The authors tried to compare their findings with different reports from worldwide. They also better justify the reason for variation among results of different research findings with respect to theirs’ based on actual situations.

Conclusions

• Normally, the conclusion emanates from the research.

6. PLOS authors have the option to publish the peer review history of their article (what does this mean?). If published, this will include your full peer review and any attached files.

Reviewer #1: No

Reviewer #2: No

---

## [Author Response · Author response to Decision Letter 0]

1 Jun 2024

Rebuttal Letter

Date: June 15, 2024

Original Manuscript ID: PONE-D-24-10619 

Original Article Title: Nasopharyngeal Carriage, Antibiotic Susceptibility Patterns, and Associated Factors of Gram-positive Bacteria among children attending the outpatient department at the University of Gondar Comprehensive Specialized Hospital, Northwest Ethiopia

To: PLOS ONE

Re: Response to academic editor and reviewers

Dear Editor,

Thank you for allowing a resubmission of our manuscript, with an opportunity to address the academic editor and reviewers’ comments.

We are uploading 

(1) Our point-to-point response to the comments below (response to academic editor and reviewer(s), respectively) 

(2) A marked-up copy with track changes (Revised Manuscript with Track Changes)

 (3) An unmarked version without track changes (Manuscript)

Best regards,

<Abebe Birhanu> et al

NOTE: We have further addressed the Journal Requirements as follows:

Concern #1. Please ensure that your manuscript meets PLOS ONE's style requirements, including those for file naming. The PLOS ONE style templates can be found at 

Authors’ response: We have adhered to PLOS ONE's style and format requirements in the revised manuscript.

Concern #2. We found significant text overlap between your submission and previous work in the (introduction, conclusion). Please revise the manuscript to rephrase the duplicated text, cite your sources, and provide details as to how the current manuscript advances on previous work. 

Authors’ response: We have revised the manuscript to address these concerns, rephrasing the duplicated text, citing sources appropriately, and detailing the advancements made by the current work.

Concern #3. Upon submission, please confirm that the manuscript, or any related manuscript, is not currently under consideration or accepted elsewhere. If related work has been submitted to PLOS ONE or elsewhere, authors must include a copy with the submitted article. Reviewers will be asked to comment on the overlap between related submissions (http://journals.plos.org/plosone/s/submission-guidelines#loc-related-manuscripts).

Authors’ response: We confirm that the manuscript, or any related manuscript, is not under consideration or accepted elsewhere. 

Concern #4. Thank you for stating the following in your Competing Interests section: 

"The authors' have no conflict of interest." Please complete your Competing Interests on the online submission form to state any Competing Interests. If you have no competing interests, please state "The authors have declared that no competing interests exist.", as detailed online in our guide for authors at http://journals.plos.org/plosone/s/submit-now. This information should be included in your cover letter; we will change the online submission form on your behalf.

Authors’ response: We have included the statement "The authors have declared that no competing interests exist" in the revised manuscript and cover letter.

Authors' Response to Additional Editor Comments

Dear Editor, we are grateful for the opportunity to revise our manuscript for PLOS ONE and for the insightful comments from the reviewers, which have significantly improved our work.

Editor Comment: Although what authors are reporting is not a novelty, they must however be commended for their efforts in reporting what is prevailing in their locality.

Authors' Response: Thank you for your positive feedback.

Editor Comment: The email addresses of authors were not needed…. except the corresponding author.

Authors' Response: We have updated the manuscript accordingly.

Editor Comment: Based on PLOS ONE authors’ guidelines, figures must be uploaded separately in TIF or another form.

Authors' Response: As requested, we have uploaded the figures in the specified format.

Editor Comment: Line 27: …It is better written as, ‘…was conducted from May 1, 2023 to August 30, 2023.’

Authors' Response: We have revised the text to "was conducted from May 1, 2023, to August 30, 2023."Please refer to line 28 in the revised manuscript.

Editor Comment: Lines 29 & 30: ‘Bacterial species were identified by biochemical tests.’ It is rewritten better as, ‘Bacterial species were identified by colony morphology, Gram stain and biochemical tests.’

Authors' Response: We have revised the text to: "Bacterial species were identified by colony morphology, Gram stain, and biochemical tests." Please refer to lines 30-31 in the revised manuscript.

Editor Comment: Lines 31 & 32: ‘…… were determined using the cefoxitin disc and D-test, respectively.’ Do you think ‘cefoxitin disc and D-test’ are equivalent terms? Drug vs. technique?

Authors' Response: We have corrected this to: "D-tests were conducted using clindamycin and erythromycin discs to detect inducible clindamycin resistance, while cefoxitin disc tests were utilized to ascertain methicillin resistance." Please refer to lines 32-34 in the revised manuscript.

Editor Comment: Lines 46: Write their long form of the ‘MRSA and MDR’

Authors' Response: We have corrected the text to "methicillin-resistant S. aureus and multi-drug resistant." Please refer to line 50 in the revised manuscript.

Editor Comment: The abstract should be rewritten and make adjustments based on the comments in the methods and results part that follows.

Authors' Response: We have revised the abstract accordingly.

Editor Comment: Once you have used the long form of the name of the bacterium then you can abbreviate the genus name throughout the body of the paper, except headings and subheadings.

Authors' Response: We have made the necessary corrections throughout the manuscript.

Editor Comment: Lines 104 to 115…. remove the objective.

Authors' Response: We have removed the objective as requested.

Editor Comment: Line 123: revise the total population of the town. It is expected above one million.

Authors' Response: We have revised the population estimate to 2,929,628. Please refer to line 122 in the revised manuscript.

Editor Comment: A separate subheading is needed as, ‘Study population’

Authors' Response: We have added a separate subheading for "Study population." Please refer to line 125 in the revised manuscript.

Editor Comment: A separate subheading for ‘Sample size determination and sampling technique’

Authors' Response: We have added a separate subheading for "Sample size determination and sampling technique." Please refer to line 131 in the revised manuscript.

Editor Comment: Remove variables of the study (lines 140-152).

Authors' Response: We have deleted the variables of the study.

Editor Comment: Why did you fail to include chocolate agar, especially for isolating fastidious organisms like S. pneumoniae?

Authors’ Response: We acknowledge that chocolate agar is a superior medium for the growth of all types of bacteria, including fastidious organisms. Blood agar, on the other hand, is also effective for cultivating both Gram-positive and Gram-negative bacteria, including some fastidious species such as S. pneumoniae and S. pyogenes. However, unlike chocolate agar, blood agar allows for the observation of hemolysis patterns, which is crucial for identifying whether S. pneumoniae is alpha, beta, or gamma hemolytic.

Editor Comment: What is the advantage of performing oxacillin and cefoxitin disc diffusion tests concurrently? Can an oxacillin susceptibility test be performed by disc diffusion for S. aureus?

Authors’ Response: Thank you for your insightful comment. Indeed, the CLSI 2022 does not recommend using the oxacillin disc diffusion method for determining the susceptibility of S. aureus. However, we used a 1µg oxacillin disc specifically to assess the non-susceptibility of S. pneumoniae to penicillin, not for S. aureus. In our study, 1µg oxacillin and 30µg cefoxitin discs were used to determine the non-susceptibility of S. pneumoniae to penicillin and to identify methicillin-resistant S. aureus, respectively. Please refer to lines 225-226 in the revised manuscript for more details.

Editor Comment: Why did you omit 'vancomycin susceptibility testing', the preferred drug for MRSA?

Authors’ Response: We appreciate this important comment. While vancomycin is indeed the last-line treatment for MRSA, the CLSI 2022 guidelines recommend using minimum inhibitory concentration (MIC) methods, such as the E-test, for vancomycin susceptibility testing in S. aureus. However, due to the unavailability of the E-test in our laboratory, we conducted antibiotic susceptibility testing using the disc diffusion method for all identified isolates in this study.

Editor Comment: Some results require both the numerator and denominator. Similarly, associated factor assessment data requires a P-value.

Authors’ Response: Thank you for your feedback. We have made the necessary corrections to include both the numerator and denominator where applicable, and we have added P-values for the assessment of associated factors.

Editor Comment: All table titles should be complete in description, area, and period of study.

Authors’ Response: We have reviewed and revised the table titles to ensure they are complete and include the description, area, and period of the study.

Editor Comment: Many of the reasons for variation discussed were already extensively covered by other authors.

Authors’ Response: We have addressed this by supplementing our discussion with additional information to explain the variations, ensuring our work contributes new insights.

Editor Comment: The conclusion was beyond your scope. Please revise it based on community/government importance, focusing on the most significant findings.

Authors’ Response: We appreciate this valuable feedback and have revised our conclusion to align more closely with our findings. We emphasize the importance of our results in raising community awareness about bacterial transmission and informing infection prevention and control strategies in hospitals.

Editor Comment: Lines 479-484: Remove authors' contributions; the system generates this automatically.

Authors’ Response: We have accepted this suggestion and removed the authors' contributions section as requested.

Editor Comment: The manuscript did not appropriately follow the PloS One writing protocol and the binomial nomenclature for bacterial names.

Authors’ Response: Thank you for highlighting this issue. We have revised the manuscript to adhere to the PloS One writing guidelines and properly use the binomial nomenclature for bacterial names.

Authors’ Response to Reviewer #1:

First, we would like to extend our gratitude to the reviewers for their critical, constructive, and timely comments. To facilitate your review, we have copied your comments below alongside our responses. We have addressed all questions and concerns point by point as follows:

Reviewer #1 Comment: The paper lacks novelty. There have been many studies in Ethiopia, including your study area, on Nasopharyngeal Carriage and Antibiotic Susceptibility Patterns of Gram-positive isolates. It would be beneficial to include additional important aspects such as genotypic characterizations of the isolates, extended study periods, and broader study groups, including inpatients.

Author’s Response: While the study may seem to lack novelty, it provides comprehensive information not covered in previous studies. For instance, previous research at the University of Gondar Comprehensive Specialized Hospital focused solely on S. pneumoniae and did not report on the inducible clindamycin resistance profiles of isolates. This study, however, highlights the overall carriage state of Gram-positive bacteria and includes their resistance profiles, including inducible clindamycin resistance. Although we stored samples at -80°C, we were unable to perform genotypic characterization due to reagent shortages. Furthermore, we excluded inpatient children for two reasons:

1. Many inpatient children are already receiving treatment, which could lead to false-negative results if we sampled them.

2. Sampling from untreated inpatient children could yield artificially high carriage rates due to the hospital environment, resulting in false-positive results.

Reviewer #1 Comment: Why focus only on Gram-positive bacteria? Gram-negative bacteria, such as Haemophilus influenzae, Klebsiella pneumoniae, and Moraxella catarrhalis, can also induce nasopharyngeal carriage.

Authors’ Response: The comment is valid, and we recognize the role of some Gram-negative bacteria in the nasopharyngeal carriage. However, due to resource limitations and the broad scope of the title, we focused our study on Gram-positive bacteria.

Reviewer #1 Comment: What about viral and fungal agents? There are fungal and viral agents that cause nasopharyngeal carriage (DOI: 10.1186/s41479-021-00088-5, https://pubmed.ncbi.nlm.nih.gov/23798081/).

Authors Response: While this is a significant comment, identifying viruses and fungi in our laboratory setting in Ethiopia is challenging due to the lack of advanced techniques and necessary reagents.

Reviewer #1 Comment: Line number 32: Please convert Epi-Data version 4.6.0.6 to 4.6.

Authors’ Response: Yes, we have accepted and corrected this. Please refer to line 35 in the revised manuscript.

Reviewer #1 Comment: Line number 43: Please convert "risk factors" to "associated factors".

Authors’ Response: Yes, we have accepted and corrected this. Please refer to line 44 in the revised manuscript.

Reviewer #1 Comment: In the abstract section, minimize the conclusions and recommendations and omit recommendations as a sub-title. Please refer to lines 49-52 in the revised manuscript.

Authors’ Response: Yes, we have accepted and corrected this.

Reviewer #1 Comment: In line number 50, please remove the colon (:) in front of the introduction.

Authors’ Response: Yes, we have accepted and corrected this. Please refer to line 54 in the revised manuscript.

Reviewer #1 Comment: The introduction section needs some modification as it focuses too much on the scientific background. It would be better to start with the problem.

Authors’ Response: We have revised the introduction to include additional information. Please refer to line 93 in the revised manuscript.

Reviewer #1 Comment: Please state the problem and prevalence of nasopharyngeal carriage and drug resistance profiles of the isolates among children in Ethiopia and the study area using figures.

Authors’ Response: Yes, we have accepted and corrected this. Please refer to lines 94-97 in the revised manuscript.

Reviewer #1 Comment: On line number 105, please convert "general objectives" to "general objective".

Authors’ Response: Yes, we have accepted and deleted this because it was also raised by the academic editor.

Reviewer #1 Comment: Why exclude children under 15 and those with a history of nasal surgery, respiratory infection, or antibiotic use from the study?

Authors’ Response: Exclusion criteria were generated to ensure accurate results. Including children with a history of nasal surgery, signs of respiratory infection, and recent antibiotic use could lead to nose obstruction, overestimated results, and underestimated results, respectively.

Reviewer #1 Comment: Please write the formula and explain how you obtained 424 and why you used a 50% proportion. There are many relevant articles, even in your study area.

Authors’ Response: We have accepted this comment and included the formula in the revised manuscript. Please refer to lines 140-150 in the revised manuscript. We used a 50% proportion for the following reasons:

1. Using the prevalence from Assefa et al., 2013, would be outdated and may not reflect the current population, leading to a low sample size.

2. The prevalence from Belayhun et al., 2023, was not used due to population and geographical differences, which would also result in a low sample size.

3. Using a 50% prevalence ensures a larger sample size, which is beneficial for statistical analysis and result interpretation.

Reviewer #1 Comment: Who analyzed the data? You mentioned that a trained laboratory technologist analyzed the data (on line numbers 155 and 156).

A

---

## [Decision Letter · Decision Letter 1]

7 Jun 2024

PONE-D-24-10619R1Nasopharyngeal Carriage, Antibiotic Susceptibility Patterns, and Associated Factors of Gram-positive Bacteria among children attending the outpatient department at the University of Gondar Comprehensive Specialized Hospital, Northwest EthiopiaPLOS ONE

Dear Dr. Birhanu,

Thank you for submitting your manuscript to PLOS ONE. After careful consideration, we feel that it has merit but does not fully meet PLOS ONE’s publication criteria as it currently stands. Therefore, we invite you to submit a revised version of the manuscript that addresses the points raised during the review process.

We look forward to receiving your revised manuscript.

Kind regards,

Tebelay Dilnessa, MSc

Academic Editor

PLOS ONE

Journal Requirements:

**Additional Editor Comments:**

See the reviwer comments and the previous comments, still they require attention.

Follow the binomical nomenclature of bacteria.

Reviewers' comments:

Reviewer's Responses to Questions

**Comments to the Author**

1. If the authors have adequately addressed your comments raised in a previous round of review and you feel that this manuscript is now acceptable for publication, you may indicate that here to bypass the “Comments to the Author” section, enter your conflict of interest statement in the “Confidential to Editor” section, and submit your "Accept" recommendation.

Reviewer #1: All comments have been addressed

Reviewer #2: All comments have been addressed

Reviewer #3: (No Response)

2. Is the manuscript technically sound, and do the data support the conclusions?

Reviewer #1: Yes

Reviewer #2: Yes

Reviewer #3: Yes

3. Has the statistical analysis been performed appropriately and rigorously? 

Reviewer #1: Yes

Reviewer #2: Yes

Reviewer #3: Yes

4. Have the authors made all data underlying the findings in their manuscript fully available?

Reviewer #1: Yes

Reviewer #2: Yes

Reviewer #3: Yes

5. Is the manuscript presented in an intelligible fashion and written in standard English?

Reviewer #1: Yes

Reviewer #2: Yes

Reviewer #3: Yes

6. Review Comments to the Author

Reviewer #1: Thanks for addressing my concerns,

As much as possible, the authors address and cover my concern.

Reviewer #2: The manuscript titled "Nasopharyngeal Carriage, Antibiotic Susceptibility Patterns, and Associated Factors of Gram-positive Bacteria among children attending the outpatient department at the University of Gondar Comprehensive Specialized Hospital, Northwest Ethiopia" presents an important study addressing the nasopharyngeal carriage and antibiotic resistance patterns of Gram-positive bacteria in a pediatric population. The study is well-structured, with clear objectives, methodology, and significant findings that contribute to the understanding of antimicrobial resistance in the region. The topic is highly relevant, particularly in the context of increasing antibiotic resistance. The findings provide valuable insights into the prevalence and resistance patterns of Gram-positive bacteria, which are crucial for informing treatment strategies and public health interventions. I have some comments here please find it. Make corrections accordingly.

Introduction: While the introduction provides a good background, it could benefit from a more detailed discussion on the global and regional implications of antibiotic resistance in Gram-positive bacteria. Including recent statistics (numerical data) on antibiotic resistance trends would strengthen the rationale for the study.

Delete the objectives part it is stated at the end of the introduction part: lines 104-115

Methods: This section should be rewritten according to the journal guidelines, as it is too wordy. Please try to condense it to align with the guidelines.

Results: The results section is well written but try to minimize it by focusing on the objectives. Table 1 should concentrate on socio-demographic characteristics to reduce its size, as the data is also available in Table 5.

Discussion: The discussion effectively interprets the findings.

Conclusion: The conclusion concisely summarizes the main findings but as much as possible please categorize your findings i.e. the prevalence data as low/medium/ high. It should also provide more specific recommendations for future research and public health policies based on your findings.

The general style of the manuscript is fine, but it does not completely follow the standard required by PLOS One Journal. Try to minimize the page of your manuscript to attract the reader by focusing on your main findings.

Reviewer #3: Please, confirm if consent was obtained for both minors and guardians, publication ethics

Please confirm any potential competing interest

7. PLOS authors have the option to publish the peer review history of their article (what does this mean?). If published, this will include your full peer review and any attached files.

Reviewer #1: No

Reviewer #2: No

Reviewer #3: No

---

## [Author Response · Author response to Decision Letter 1]

2 Jul 2024

Rebuttal Letter

Date: July 2, 2024

Original Manuscript Number: PONE-D-24-10619R1

Original Article Title: Nasopharyngeal Carriage, Antibiotic Susceptibility Patterns, and Associated Factors of Gram-positive Bacteria among children attending the outpatient department at the University of Gondar Comprehensive Specialized Hospital, Northwest Ethiopia

To: PLOS ONE

Re: Response to academic editor and reviewers

Dear Editor,

Thank you for allowing a resubmission of our manuscript, with an opportunity to address the academic editor and reviewers’ comments.

We are uploading 

(1) Our point-by-point response to the comments below (response to academic editor and reviewer(s), respectively) 

(2) A marked-up copy with track changes (Revised Manuscript with Track Changes)

 (3) An unmarked version without track changes (Manuscript)

Best regards,

<Abebe Birhanu> et al

NOTE: We have further addressed the Journal Requirements as follows:

Concern #1. Please review your reference list to ensure that it is complete and correct. If you have cited papers that have been retracted, please include the rationale for doing so in the manuscript text, or remove these references and replace them with relevant current references. Any changes to the reference list should be mentioned in the rebuttal letter that accompanies your revised manuscript. If you need to cite a retracted article, indicate the article’s retracted status in the References list and also include a citation and full reference for the retraction notice.

Authors’ Response: We have thoroughly reviewed each reference and have not included any retracted articles.

Authors' Response to Additional Editor Comments

Dear Editor, we are grateful for the opportunity to revise our manuscript again for PLOS ONE and for the insightful comments from the reviewers, which have significantly improved our work.

Editor Comment: See the reviewer comments and the previous comments, still they require attention. Follow the binomial nomenclature of bacteria.

Authors’ Response: We thank you for highlighting this issue again. We have revised the manuscript to properly use the binomial nomenclature for bacterial names.

Authors’ Response to Reviewer #1:

First, we would like to extend our gratitude to the reviewers for their critical, constructive, and timely comments. To facilitate your review, we have copied your comments below alongside our responses. We have addressed all questions and concerns point by point as follows:

Reviewer #1 Comment: Thanks for addressing my concerns. As much as possible, the authors address and cover my concerns.

Authors’ Response: We thank you for providing good feedback.

Authors’ response to Reviewer #2:

Reviewer #2 Comment: The manuscript titled "Nasopharyngeal Carriage, Antibiotic Susceptibility Patterns, and Associated Factors of Gram-positive Bacteria among children attending the outpatient department at the University of Gondar Comprehensive Specialized Hospital, Northwest Ethiopia" presents a significant study addressing nasopharyngeal carriage and antibiotic resistance patterns in Gram-positive bacteria within a pediatric population. The study is well-organized with clear objectives, methodology, and findings that are valuable for understanding antimicrobial resistance in the region. The topic is highly pertinent given the rising concern of antibiotic resistance. The results offer important insights into the prevalence and resistance patterns of Gram-positive bacteria, crucial for guiding treatment strategies and public health measures.

Authors’ Response: We thank you for giving positive feedback.

Reviewer #2 Comment: While the introduction provides a solid background, it would benefit from a more comprehensive discussion on the global and regional implications of antibiotic resistance in Gram-positive bacteria. Including recent numerical data on antibiotic resistance trends would enhance the study's rationale.

Authors’ Response: We thank you for raising this issue. We have included the additional information in the introduction section as requested in the revised manuscript. Please see lines 108-113 of the manuscript.

Reviewer #2 Comment: Remove the objectives section stated at the end of the introduction: lines 104-115.

Authors’ Response: We have removed the objectives section from the end of the introduction as requested in the revised manuscript.

Reviewer #2 Comment: The “Methods” section is too lengthy and should be rewritten to conform to the journal guidelines. Please condense it accordingly.

Authors’ Response: We have revised and condensed the methods section to align with the journal guidelines.

Reviewer #2 Comment: The results section is well-written but should be minimized by focusing on the objectives. Table 1 size should be reduced to focus on socio-demographic characteristics, as this data is also presented in Table 5.

Authors’ Response: We have revised the results section to better align with the objectives and have minimized the size of Table 1 accordingly. Please refer to lines 261-268 of the manuscript.

Reviewer #2 Comment: The discussion effectively interprets the findings.

Authors’ Response: We thank you for providing good feedback.

Reviewer #2 Comment: The conclusion succinctly summarizes the main findings. However, please categorize your findings (e.g., prevalence data as low/medium/high) and provide more specific recommendations for future research and public health policies based on your findings.

Authors’ Response: We thank you for the suggestion. We have categorized the findings accordingly and added specific recommendations in the revised manuscript. Please see lines 470-482 of the manuscript.

Reviewer #2 Comment: The overall style of the manuscript is acceptable but does not fully comply with the PLOS One Journal standards. Please try to minimize the page of the manuscript by focusing on the main findings.

Authors’ Response: We have minimized the manuscript’s length and made the required corrections to adhere to the PLOS ONE Journal standards.

Reviewer #3 Comment: Confirm if consent was obtained from both minors and guardians, regarding publication ethics.

Authors’ Response: We confirmed that written informed assent and consent were secured from both the children and their guardians.

Reviewer #3 Comment: Confirm any potential competing interests.

Authors’ Response: We thank you for raising this issue again. The authors have declared that no competing interests exist and have approved the final submission of the revised manuscript.

Title: - Nasopharyngeal Carriage, Antibiotic Susceptibility Patterns, and Associated Factors of Gram-positive Bacteria among children attending the outpatient department at the University of Gondar Comprehensive Specialized Hospital, Northwest Ethiopia

Abstract

Comment #1: Emphasize antibiotic resistance findings: Highlight the significance of the methicillin resistance observed in S. aureus and coagulase-negative staphylococci. Strengthen the conclusion, summarize the main findings succinctly, and highlight their implications for public health. Highlight the practical implications of the findings and how they can inform clinical practices, policy decisions, and future research directions.

Authors’ Response: We thank you for pointing out the comment and we have made corrections accordingly. The significance of methicillin resistance observed in both S. aureus and coagulase-negative staphylococci has the following implications for public health, clinical practice, and future directions:

1. Methicillin-resistant S. aureus (MRSA) and methicillin-resistant coagulase-negative staphylococci (MRCoNS) are resistant to beta-lactam antibiotics, including methicillin and other penicillin derivatives. This resistance complicates treatment options for infections caused by these bacteria, as they are often resistant to multiple classes of antibiotics, limiting effective therapeutic choices.

2. MRSA and MRCoNS are associated with higher morbidity and mortality rates compared to methicillin-sensitive strains. They are often responsible for healthcare-associated infections (HAIs) and community-acquired infections, posing a significant burden on healthcare systems and increasing healthcare costs.

3. The presence of methicillin resistance underscores the importance of antibiotic stewardship programs to ensure appropriate use of antibiotics, minimize resistance development, and preserve the effectiveness of available antimicrobial agents.

4. MRSA and MRCoNS are notorious for their ability to spread easily in healthcare settings, leading to outbreaks. Effective infection control measures, including hand hygiene, isolation protocols, and environmental cleaning, are crucial to prevent transmission.

5. Continuous monitoring of methicillin resistance patterns in S. aureus and coagulase-negative staphylococci is essential for understanding epidemiological trends, guiding treatment strategies, and developing new therapeutic approaches, such as novel antibiotics or alternative treatment modalities.

Study design, period, and area 

Comment #2: Include additional information about the hospital such as its specialty areas, facilities, and departments relevant to the study.

Authors’ Response: We have added additional information “The UoGCSH, a major teaching hospital with 977 beds, and 29 wards and emergency rooms, has served nearly 13 million people in its surrounding area and neighboring regions. It provides a wide range of medical services, including internal medicine, surgery, obstetrics and gynecology, pediatrics, laboratory tests, eye care, physiotherapy, dental care, cervical health, psychiatry, dermatology, and drug supply. The hospital also offers various social services and has specialized units for tuberculosis, kala-azar, cancer treatment, fistula surgery, psychiatric and psychological treatment, palliative and rehabilitation services, adult and pediatric intensive care unit, and cataract surgery” to the revised manuscript. We have attached source files as references in the “Other file type” section of the system. Please see it accordingly.

Comment #3: Clarify the population size estimation: Explain the methodology used to estimate the total population of Gondar town as 2,929,628. Provide references or sources for this estimation to establish its validity.

Authors’ Response: We appreciate your concern and have taken steps to address it by gathering updated information from local sources. As per your request, we have obtained information from the Central Statistical Agency of Ethiopia, Gondar branch, indicating that the estimated population of Gondar town in 2023 is 487,224. This estimation is obtained based on population projection methods utilizing data from the 2007 census data, as Ethiopia has not conducted a new census since that time. For detailed verification, we have attached supporting documents received from the Central Statistical Agency of Ethiopia, Gondar branch as references in the “Other files type” section of the system. Please refer to these files for further information. 

Study population 

Comment #4: Explain why these departments were chosen for the study and how they are representative of the target population.

Authors’ Response: We appreciate you for bringing this issue to our minds. Outpatient children were chosen for the study of nasopharyngeal carriage of Gram-positive bacteria for several reasons, making them representative of the target population effectively:

1. Outpatient children often visit clinics for respiratory infections, which makes them a suitable group for studying nasopharyngeal bacterial carriage. These infections are common in the general pediatric population, ensuring that the study captures a relevant and typical scenario.

2. Outpatient settings provide easier access to a diverse and large number of children compared to inpatient settings. This makes it practical to recruit participants and conduct the study without the constraints associated with hospitalized children, such as severe illness or hospitalization-related factors.

3. Outpatients generally represent less severe cases of illness compared to inpatients. This is crucial for studying nasopharyngeal carriage because it reflects bacterial colonization in a typical health condition rather than an extreme one, offering a more accurate picture of the bacteria present in the general child population.

4. Choosing outpatient children helps minimize selection bias. Hospitalized children might have been exposed to hospital-acquired infections or may have underlying conditions that could skew the results. Outpatients, on the other hand, are more likely to represent a healthy, broader pediatric population.

5. Outpatient clinics see children from a wide range of demographics, including different ages, socio-economic backgrounds, and health statuses. This diversity makes the sample more representative of the overall child population.

6. Outpatients typically live in the community and are thus more representative of the bacteria circulating in the community than those specific to hospital environments.

Specimen collection, transportation, and processing

Comment #5: Explain the specific responsibilities of the trained laboratory technologist in data collection. Elaborate on their training and expertise to establish their credibility and reliability in accurately collecting the data.

Authors’ Response: We thank you for pointing out this issue again. By "trained laboratory technologists," we refer to professionals who hold BSc degrees in medical laboratory sciences and MSc degrees in Medical Microbiology. In addition, they have extensive experience working in bacteriology laboratories at the University of Gondar Comprehensive Specialized Hospital. They are also certified by the Ethiopian Public Health Institute in standard bacteriology procedures and have experience as data collectors for many PhD and MSc research projects. In our study, these technologists were responsible for accurately labeling patient information, transporting samples to the laboratory under conditions that prevent degradation or contamination, and correctly interpreting and recording laboratory test results.

Comment #6: Mention any quality control measures taken during the data collection process.

Authors’ Response: We thank you for highlighting this concern. We understand that quality control is crucial for ensuring the accuracy, reliability, and validity of study results. In our study on the nasopharyngeal carriage of Gram-positive bacteria, we implemented the following quality control measures:

1. Trained nurses collected nasopharyngeal samples using sterile nasopharyngeal swabs and standardized procedures to ensure the outcome of the results.

2. A rigorous labeling system was used to accurately identify and trace each sample with the corresponding study participants throughout the study, with appropriate transport media and specified temperature maintenance to preserve sample integrity.

3. Standard bacterial reference strains were received from the Ethiopian Public Health Institute to verify the accuracy and reliability of the test results.

4. Double data entry was performed by different individuals, with any discrepancies resolved to ensure data accuracy.

5. Informed consent was obtained from all participants or their guardians, adhering to ethical guidelines.

Results 

Comment #7: Provide more context for previous hospital visits, respiratory tract infections, and antibiotic use: Expand on the significance of these factors in the study. Explain why previous hospital visits, respiratory tract infections, and antibiotic use within the last three months were relevant to the research question or hypothesis. 

Authors’ Response: We appreciate you for raising such an important comment. In a study on the nasopharyngeal carriage of Gram-positive bacteria in outpatient children, examining previous hospital visits, recent respiratory tract infections, and antibiotic use is highly significant for several reasons. These factors can influence the prevalence and types of bacteria present in the nasopharynx and help contextualize the study's findings. Here’s a more detailed explanation of why these factors are relevant to the research question or

---

## [Decision Letter · Decision Letter 2]

4 Jul 2024

PONE-D-24-10619R2Nasopharyngeal Carriage, Antibiotic Susceptibility Patterns, and Associated Factors of Gram-positive Bacteria among children attending the outpatient department at the University of Gondar Comprehensive Specialized Hospital, Northwest EthiopiaPLOS ONE

Dear Dr. Birhanu,

Thank you for submitting your manuscript to PLOS ONE. After careful consideration, we feel that it has merit but does not fully meet PLOS ONE’s publication criteria as it currently stands. Therefore, we invite you to submit a revised version of the manuscript that addresses the points raised during the review process.

We look forward to receiving your revised manuscript.

Kind regards,

Tebelay Dilnessa, MSc

Academic Editor

PLOS ONE

Journal Requirements:

Additional Editor Comments:The paper was significantly improved, still requires a critical revision and proofreading.Lines 40-43: Correct the following as follows: *Staphylococcus aureus* was the most prevalent 122/424 (28.8%), followed by *Streptococcus pneumoniae* 92/424 (21.7%). Methicillin resistance was observed in 19/122 (15.6%) of *S. aureus* and 3/60 (5%) of coagulase-negative staphylococcus (CoNS) species. Inducible clindamycin resistance was 10/122 (8.2%) in *S. aureus* and 4/53 (7.5%) in coagulase-negative staphylococcus species.Italics the ‘P’ to refer p-value throughout the document.The conclusion part of the abstract should contain concrete ideas and should emanate from the result based on your objective.Lines 63-65: It is better written as, …………. *S. pneumoniae, S. aureus*, coagulase-negative staphylococcus species (CoNS), and *Streptococcus pyogenes* [2, 3].Line 84 and 88: Write as, Coagulase-negative staphylococcus species…………..Line 119 and 120 :……..…. this study addressed gaps left by prior research by offering data on the carriage of Gram-positive bacteria and their antimicrobial resistance patterns, including inducible……….It is preferable to use ‘antimicrobial’ than ‘drug’. For example, line 122 throughout the document. Uniformity is more important.Unnecessary space between the heading and subheading should be removed.Line 125: The subheading ‘Study design, period, and area’. I recommend you to split into two subheadings as ‘Study area and setting’ and ‘Study design and period’Line 135: Gondar town is located 750 km……..Line 147: ‘Concisely’ was not the correct word.Line 168: Specimen collection and processingLine 181, 184: Write the source of the media (manufacturer/city/country) similar to line 191.Line 192: …….Gram-positive diplococci…..How could you standardize the suspension preparation during performing antimicrobial susceptibility testing?In the methods and result part, try to shorten the subheadings. For example, line 272, 279 and 299 can be written as follows, respectively: ‘Prevalence of Gram-positive bacteria’, ‘Antimicrobial susceptibility pattern’ & ‘Inducible clindamycin resistance’Line 208: Antimicrobial susceptibility testingLine 227 and 228: ……… Results of antimicrobial susceptibility testing………………….Line 230: ………….. multidrug resistance (MDR) [ 2]. Additionally, the reference you mentioned (reference 2) for MDR was not the correct reference/ and cannot be the source. It is a cross-reference and replace it by the correct reference.Line 257: A letter of permission was obtained from……………Line 309: Multi-drug resistance patternLine 324: Associated factors for nasopharyngeal carriage421: The MDR proportion of 69.2%, 40%, 39.1%, and 18.2% were found in the current study for *S. aureus*,……..The conclusion and the recommendation were beyond your scope, so make it specific to your study.All tables and figure (s) should be cited in the text, otherwise it/they should be removed.Figure 1 should be prepared without description in tif and uploaded. The description should be placed within the text where you cited.

Reviewers' comments:

Reviewer's Responses to Questions

**Comments to the Author**

1. If the authors have adequately addressed your comments raised in a previous round of review and you feel that this manuscript is now acceptable for publication, you may indicate that here to bypass the “Comments to the Author” section, enter your conflict of interest statement in the “Confidential to Editor” section, and submit your "Accept" recommendation.

Reviewer #2: All comments have been addressed

2. Is the manuscript technically sound, and do the data support the conclusions?

Reviewer #2: Yes

3. Has the statistical analysis been performed appropriately and rigorously? 

Reviewer #2: Yes

4. Have the authors made all data underlying the findings in their manuscript fully available?

Reviewer #2: Yes

5. Is the manuscript presented in an intelligible fashion and written in standard English?

Reviewer #2: Yes

6. Review Comments to the Author

Reviewer #2: The study is well-organized with clear objectives, methodology, and findings that are valuable for understanding antimicrobial resistance in the region. The topic is highly pertinent given the rising concern of antibiotic resistance. The results offer important insights into the prevalence and resistance patterns of Gram-positive bacteria, crucial for guiding treatment strategies and public health measures. The authors addressed all my concerns effectively. Thank you.

7. PLOS authors have the option to publish the peer review history of their article (what does this mean?). If published, this will include your full peer review and any attached files.

Reviewer #2: No

---

## [Author Response · Author response to Decision Letter 2]

13 Jul 2024

Rebuttal Letter

Date: July 13, 2024

Original Manuscript Number: PONE-D-24-10619R2

Original Article Title: Nasopharyngeal Carriage, Antibiotic Susceptibility Patterns, and Associated Factors of Gram-positive Bacteria among children attending the outpatient department at the University of Gondar Comprehensive Specialized Hospital, Northwest Ethiopia

To: PLOS ONE

Re: Response to academic editor and reviewers

Dear Editor,

Thank you for giving us the chance to resubmit our manuscript and respond to the comments from the academic editor and reviewers.

We are uploading 

(1) Our point-by-point response to academic editor and reviewer(s) comments below (Response to Reviewers).

(2) A marked-up copy with track changes (Revised Manuscript with Track Changes)

 (3) An unmarked version without track changes (Manuscript)

Best regards,

<Abebe Birhanu> et al

NOTE: We have further addressed the Journal Requirements as follows:

Concern #1. Please review your reference list to ensure that it is complete and correct. If you have cited papers that have been retracted, please include the rationale for doing so in the manuscript text, or remove these references and replace them with relevant current references. Any changes to the reference list should be mentioned in the rebuttal letter that accompanies your revised manuscript. If you need to cite a retracted article, indicate the article’s retracted status in the References list and also include a citation and full reference for the retraction notice.

Authors’ Response: Thank you for this valuable comment. We have entirely reviewed each reference and properly cited them in the manuscript. 

Authors' Response to Additional Editor Comments

Dear Editor, we are grateful for the opportunity to revise our manuscript again for PLOS ONE and for the insightful comments from the reviewers, which have significantly improved our work. 

Editor Comment: The paper was significantly improved, still requires a critical revision and proofreading.

Authors’ Response: Thank you for your feedback. We have critically revised the paper accordingly. 

Editor Comment: Lines 40-43: Correct the following as follows: Staphylococcus aureus was the most prevalent 122/424 (28.8%), followed by Streptococcus pneumoniae 92/424 (21.7%). Methicillin resistance was observed in 19/122 (15.6%) of S. aureus and 3/60 (5%) of coagulase-negative staphylococcus (CoNS) species. Inducible clindamycin resistance was 10/122 (8.2%) in S. aureus and 4/53 (7.5%) in coagulase-negative staphylococcus species. 

Authors’ Response: Thank you for the correction. We have made the changes as requested. Please refer to lines 40-44.

Editor Comment: Italics the ‘P’ to refer p-value throughout the document.

Authors’ Response: Thank you for this important comment. We have italicized the letter "P" throughout the manuscript.

Editor Comment: The conclusion part of the abstract should contain concrete ideas and should emanate from the result based on your objective.

Authors’ Response: Thank you for bringing this to our attention. We have revised the abstract to include concrete ideas based on our objectives. Please refer to lines 50-61.

Editor Comment: Lines 63-65: It is better written as, …………. S. pneumoniae, S. aureus, coagulase-negative staphylococcus species (CoNS), and Streptococcus pyogenes.

Authors’ Response: We have accepted and corrected accordingly. Please check lines 66-67.

Editor Comment: Line 84 and 88: Write as, Coagulase-negative staphylococcus species………….

Authors’ Response: We have made the corrections as suggested. Please refer to lines 86 and 90.

Editor Comment: Line 119 and 120….…. this study addressed gaps left by prior research by offering data on the carriage of Gram-positive bacteria and their antimicrobial resistance patterns, including inducible……….

Authors’ Response: We have replaced "address" with "addressed" and "antibiotic" with "antimicrobial." Please see lines 122-123.

Editor Comment: It is preferable to use ‘antimicrobial’ than ‘drug’. For example, line 122 throughout the document. Uniformity is more important.

Authors’ Response: We have replaced the word "drug" with "antimicrobial" throughout the manuscript. Please refer to line 125.

Editor Comment: Unnecessary space between the heading and subheading should be removed.

Authors’ Response: Thank you for highlighting this. We have removed the unnecessary spaces between headings and subheadings.

Editor Comment: Line 125: The subheading ‘Study design, period, and area’. I recommend you to split into two subheadings as ‘Study area and setting’ and ‘Study design and period’

Authors’ Response: We have made the suggested changes. Please check lines 128 and 139.

Editor Comment: Line 135: Gondar town is located 750 km…….

Authors’ Response: We have added the verb "is" to the text. Please refer to line 137.

Editor Comment: Line 147: ‘Concisely’ was not the correct word.

Authors’ Response: We have changed "Concisely" to "Briefly" in the text. Please check line 151.

Editor Comment: Line 168: Specimen collection and processing

Authors’ Response: We have removed the word "transportation" as requested. Please see line 172.

Editor Comment: Line 181, 184: Write the source of the media (manufacturer/city/country) similar to line 191.

Authors’ Response: We have added the manufacturer and country "HiMedia, India" to the manuscript. Please check lines 185 and 188.

Editor Comment: Line 192: ……. Gram-positive diplococci….

Authors’ Response: We have corrected "Gram-positive cocci" to "Gram-positive diplococci" in the text. Please see line 191.

Editor Comment: How could you standardize the suspension preparation during performing antimicrobial susceptibility testing?

Authors’ Response: We appreciate this valuable comment. We have added the following to the revised manuscript: "To standardize the bacterial suspension preparation, 0.85% normal saline was used. If the suspension was more turbid than the 0.5 McFarland standard, it was diluted accordingly. Conversely, if the suspension was less turbid, additional pure colonies of the target bacteria were added to achieve the correct turbidity." Please refer to lines 215-218 in the revised manuscript.

Editor Comment: In the methods and result part, try to shorten the subheadings. For example, line 272, 279, and 299 can be written as follows, respectively: ‘Prevalence of Gram-positive bacteria’, ‘Antimicrobial susceptibility pattern’ & ‘Inducible clindamycin resistance’

Authors’ Response: We have made the requested changes. Please refer to lines 278, 287, and 306.

Editor Comment: Line 208: Antimicrobial susceptibility testing

Authors’ Response: We have corrected it as requested. Please refer to line 213.

Editor Comment: Line 227 and 228: ……… Results of antimicrobial susceptibility testing………………….

Authors’ Response: We have made the necessary corrections by replacing “antibiotic” with “antimicrobial”. Please see lines 233-234.

Editor Comment: Line 230: …………. multidrug resistance (MDR) [ 2]. Additionally, the reference you mentioned (reference 2) for MDR was not the correct reference/ and cannot be the source. It is a cross-reference and replace it by the correct reference.

Authors’ Response: Thank you for pointing this out. We have replaced "reference 2" with the correct reference (Magiorakos et al., 2012). Please refer to line 236.

Editor Comment: Line 257: A letter of permission was obtained from……………

Authors’ Response: We have replaced the word "acquired" with "obtained." Please check line 263. 

Editor Comment: Line 309: Multi-drug resistance pattern

Authors’ Response: We have corrected it accordingly. Please see line 316.

Editor Comment: Line 324: Associated factors for nasopharyngeal carriage

Authors’ Response: We have corrected it as requested. Please check line 332.

Editor Comment: 421: The MDR proportion of 69.2%, 40%, 39.1%, and 18.2% were found in the current study for S. aureus…….

Authors’ Response: Thank you for the suggestion. We have revised it to: "The MDR proportions of 69.2%, 40%, 39.1%, and 18.2% were found in the current study for S. aureus, CoNS, S. pneumoniae, and S. pyogenes, respectively." Please refer to line 429.

Editor Comment: The conclusion and the recommendation were beyond your scope, so make it specific to your study.

Authors’ Response: We have revised the conclusion and recommendations to align with our findings and objectives. Please check lines 482-497.

Editor Comment: All tables and figure (s) should be cited in the text, otherwise it/they should be removed.

Authors’ Response: We have cited all tables and figures in the text at the end of their respective sentences. Please refer to lines 299, 337, and 347.

Editor Comment: Figure 1 should be prepared without description in tif and uploaded. The description should be placed within the text where you cited.

Authors’ Response: We have uploaded Figure 1 without description in "tif" format in the "figure section" of the system, and the description is placed within the text where it is cited. Please see lines 283-286.

Authors’ Response to Reviewer #2:

First, we would like to extend our gratitude to the reviewers for their critical, constructive, and timely comments. To facilitate your review, we have copied your comments below alongside our responses. We have addressed all questions and concerns point by point as follows:

Reviewer #2 Comment: The study is well-organized with clear objectives, methodology, and findings that are valuable for understanding antimicrobial resistance in the region. The topic is highly pertinent given the rising concern of antibiotic resistance. The results offer important insights into the prevalence and resistance patterns of Gram-positive bacteria, crucial for guiding treatment strategies and public health measures. The authors addressed all my concerns effectively. Thank you.

Authors’ Response: We appreciate your positive feedback. Thank you.

---

## [Editor Report · Decision Letter 3]

16 Jul 2024

Nasopharyngeal Carriage, Antimicrobial Susceptibility Patterns, and Associated Factors of Gram-positive Bacteria among children attending the outpatient department at the University of Gondar Comprehensive Specialized Hospital, Northwest Ethiopia

PONE-D-24-10619R3

Dear Dr. Birhanu,

We’re pleased to inform you that your manuscript has been judged scientifically suitable for publication and will be formally accepted for publication once it meets all outstanding technical requirements.

Kind regards,

Tebelay Dilnessa, MSc

Academic Editor

PLOS ONE
---

## [Editor Report · Acceptance letter]

18 Jul 2024

PONE-D-24-10619R3 

PLOS ONE

Dear Dr. Birhanu, 

I'm pleased to inform you that your manuscript has been deemed suitable for publication in PLOS ONE. Congratulations! Your manuscript is now being handed over to our production team.

Kind regards, 

on behalf of

Dr. Tebelay Dilnessa 

Academic Editor

PLOS ONE